# Functional architecture of executive control and associated event-related potentials in macaques

Amirsaman Sajad[1,3], Steven P. Errington [1,3] & Jeffrey D. Schall [1,2] ✉

The medial frontal cortex (MFC) enables executive control by monitoring relevant information and using it to adapt behavior. In macaques performing a saccade countermanding (stop-signal) task, we simultaneously recorded electrical potentials over MFC and neural spiking across all layers of the supplementary eye field (SEF). We report the laminar organization of neurons enabling executive control by monitoring the conflict between incompatible responses, the timing of events, and sustaining goal maintenance. These neurons were a mix of narrow-spiking and broad-spiking found in all layers, but those predicting the duration of control and sustaining the task goal until the release of operant control were more commonly narrow-spiking neurons confined to layers 2 and 3 (L2/3). We complement these results with evidence for a monkey homolog of the N2/P3 event-related potential (ERP) complex associated with response inhibition. N2 polarization varied with error-likelihood and P3 polarization varied with the duration of expected control. The amplitude of the N2 and P3 were predicted by the spike rate of different classes of neurons located in L2/3 but not L5/6. These findings reveal features of the cortical microcircuitry supporting executive control and producing associated ERPs.

Effective control of behavior is necessary to override conflicting, habitual, or inappropriate responses, and to facilitate stopping, switching, and updating of task goals. Investigating features of executive control is afforded through the countermanding (stop-signal) task[1], during which macaque monkeys, like humans, exert response inhibition and adapt performance based on stimulus history, response outcomes, and the temporal structure of task events[2].

Converging evidence from imaging, electrophysiology, and lesion studies indicates that MFC, including the supplementary motor complex, is essential for executive control[3–6]. In humans, noninvasive ERP measures derived from a negative–positive waveform over the medial frontal cortex, known as the N2/P3, have been used to test hypotheses about executive control function[7]. Executive control of gaze is mediated by the SEF—an agranular area on the dorsomedial convexity of the MFC. Here, neurons modulate too late to enable reactive control of eye

movements and instead contribute to proactive control[8]. Electrical microstimulation of SEF improves performance in the countermanding task by slowing response time (RT)[9] by postponing the accumulation of pre-saccadic activity in gaze control structures[10]. SEF also supports working memory[11,12] and signals surprising events[13], event timing[14,15], response conflict[16], and consequences[17].

Whilst these executive control signals are well documented in MFC, how they arise is uncertain[3–6]. Understanding neural spiking with laminar resolution is necessary to clarify circuit-level mechanisms because neurons in different layers have different biophysical properties and anatomical connections. Although such approaches have been integral in developing our understanding of processing in the early visual system[18] the canonical cortical microcircuit derived from granular sensory areas[19] does not explain agranular frontal areas like SEF[20–24].

[1]Department of Psychology, Vanderbilt Vision Research Center, Center for Integrative & Cognitive Neuroscience, Vanderbilt University, Nashville, TN, USA. [2]Department of Biology, Centre for Vision Research, Vision Science to Application, York University, Toronto, ON, Canada. [3]These authors contributed equally: Amirsaman Sajad, Steven P. Errington. ✉e-mail: schalljd@yorku.ca

To understand mechanistically how executive control signals are generated in the MFC, we sampled neural spiking activity in SEF beneath where the frontal N2/P3 is sampled. We previously described the laminar microcircuitry of performance monitoring signals in the SEF and their relationship to error-related negativity (ERN)[17]. Here we describe the laminar microcircuitry of signals that monitor events occurring during successful stopping performance. We describe three classes of neurons that signal response conflict, event timing, and maintenance of task goals. We also provide evidence that macaque monkeys produce the N2/P3 ERP associated with response inhibition, describe the task factors indexed by this ERP, and elucidate the neuron classes predicting the polarization.

## Results

### Countermanding performance and neural sampling

Neurophysiological and electrophysiological data were recorded from two macaque monkeys performing the saccade stop-signal task with explicit feedback tone cues preceding the possible delivery of fluid reward (Fig. 1a)[25]. Data collection and analysis were informed by the consensus guide for the stop-signal task[1]. Briefly, monkeys earned fluid reward for shifting their gaze to a target on its appearance and for inhibiting the planned saccade when an infrequent stop-signal appeared. The delay of the stop-signal was varied experimentally to yield an equal probability of successful (canceled) or unsuccessful (noncanceled, NC) stop-signal trials. Two kinds of NC trials were distinguished (Fig. 1a): NC trials in which the gaze shifted after the stop-signal appeared (RT > SSD) were identified as errors (NC$_{error}$), whereas trials when gaze shifted before SSD (RT < SSD) were identified as premature noncanceled trials (NC$_{premature}$). Both trials resulted in no juice reward being delivered.

We acquired 33,816 trials across 29 sessions (Monkey Eu, male, 12 sessions, 11,583 trials; Monkey X, female, 17 sessions, 22,233 trials). Typical performance was produced by both monkeys. First, response times (RT) on noncanceled trials (mean ± SD Eu: 294 ± 179 ms; X:

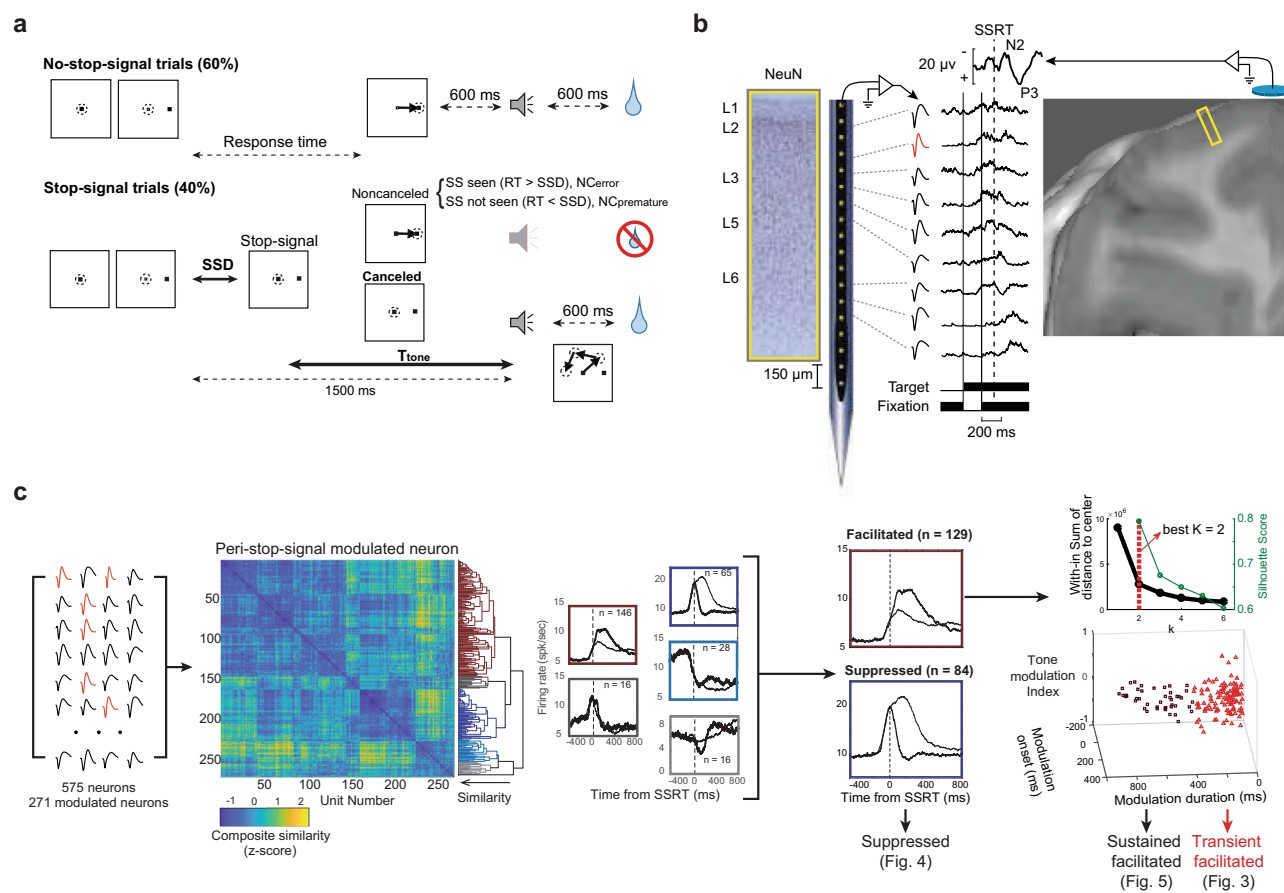

**Fig. 1 | Experimental approach. a** Saccade countermanding task. Monkeys earned fluid reward for shifting gaze (dashed circle) to a visual target unless a stop-signal appeared after a variable stop-signal delay (SSD) adjusted to achieve ~50% canceled trials. Successful no-stop-signal or canceled trial outcome was signaled by a high-pitched tone after $T_{tone}$ preceding fluid delivery. Noncanceled trials were signaled by a low-pitched tone. Monkeys could shift gaze and blink after $T_{tone}$, which equaled 1500 ms - SSD. Noncanceled trials with RT > SSD were explicit errors. Noncanceled trials with RT < SSD were premature responses. Details in text. **b** Neural sampling. Neural spiking was recorded across all layers of agranular SEF (NeuN stain) using Plexon U-probe. Neurons with both broad (black) and narrow (red) spikes were sampled. Spiking modulation was measured relative to the presentation of task events (thin solid, visual target; thick solid, stop-signal) and performance measures like SSRT (dashed vertical). Simultaneously, EEG was recorded from an electrode in the cranial surface over the MFC (10–20 location Fz). The yellow rectangle portrays the cortical area sampled in a T1 MR image. **c** Neuron classification. Among 575 sampled neurons 271 were classified by their modulation during successful stopping. Their respective spike density functions (SDF) were submitted to an unsupervised consensus clustering pipeline[27] yielding the composite similarity matrix (z-score color map) and associated dendrogram. The color coding in this figure bears no relation to colors used in other figures. Consensus clustering yielded 5 clusters with 2 clusters containing >80% of the neurons. The different modulation patterns are evident in the average SDF aligned on SSRT of each cluster. Following manual curation, 129 facilitated and 84 suppressed neurons were analyzed further based on heterogeneity in modulation latency, duration, and pattern with the feedback tone. K-means clustering further divided facilitated neurons into sustained and transient classes (right panel). The inset in the right panel shows that k = 2 clusters were chosen based on the Elbow method and the Silhouette score. Therefore, three classes of neurons were analyzed in this study.

230 ± 83 ms) were systematically shorter than those on no-stop-signal-signal trials (Eu: 313 ± 119 ms, X: 263 ± 112 ms; Mixed-effects linear regression (two-tailed) grouped by monkey, $t(27507) = -17.4$, $p < 0.001$). Second, the probability of noncanceled trials increased with stop-signal delay (SSD). These two observations validated the use of the independent race model[26] to estimate the stop-signal reaction time (SSRT), an estimate of the time needed to cancel a partially pre-pared saccade. SSRT measures the time of successful stopping. If the stop process does not inhibit saccade preparation before this time, then gaze will shift. Accordingly, neural modulation occurring before SSRT can contribute to reactive stopping but modulation occurring after SSRT cannot[8,27]. SSRT across sessions did not differ between monkeys (Eu: 118 ± 23 ms, X: 103 ± 24 ms, independent groups $t$-test (two-tailed), $t(27) = -1.69$, $p = 0.103$).

Electroencephalogram (EEG) was recorded over the MFC with leads placed on the cranial surface beside the chamber while a linear electrode array (Plexon, 24 channels, 150 μm spacing) was inserted in SEF (Fig. 1b). SEF was localized by anatomical landmarks and intra-cortical electrical microstimulation[21]. Neural spiking was sampled from 575 single units (Eu: 244, X: 331) across five sites. Overall, 213 neurons (Eu: 105, X: 108) modulated around SSRT and revealed the functional signals reported in this study. The description of the laminar dis-tribution of signals is based on 16 of the 29 sessions during which electrode arrays were oriented perpendicular to the cortical layers. In total, 119 neurons (Eu: 54, X: 65) were assigned to layers confidently (Supplementary Fig. 1b, Supplementary Table 1). Due to variability in the estimates and the indistinct border between L6 and white matter, units appearing beneath the average gray-matter estimate were assigned to L6.

## Functional classification of neural activity related to successful stopping

Aspects of the behavioral and neural dataset analyzed here have been used to address other questions. The current results describe the activity on successfully canceled stop-signal trials. According to the race model[26] and validated neurophysiologically[27] noncanceled trials result from faster responses than canceled trials (RT < SSD + SSRT, Supplementary Fig. 2a) and therefore are associated with distinct neural mechanisms. Activity on noncanceled stop-signal trials was described previously[17].

We classified neurons primarily based on the pattern of neural activation around SSRT (Fig. 1c, Supplementary Fig. 1c). A semi-supervised consensus cluster algorithm[28] followed by manual curation revealed populations of neurons that were facilitated (129) or sup-pressed (84) following successful response inhibition. Facilitated neurons were further classified into two distinct subpopulations based on other response profile properties (Fig. 1c).

To identify the putative functional roles of each pattern of mod-ulation, we contrasted neural modulation on canceled trials with a subset of no-stop-signal trials with matching temporal dynamics, identified through a process of latency-matching. This process only included no-stop-signal trials with response times long enough to have been canceled had the stop-signal been presented (RT ≥ SSD + SSRT; Supplementary Fig. 2a). We distinguished canceled trials by SSD to account for variations in modulation dynamics arising from differ-ences in the timing of task events (Fig. 1a). We also examined the neural activity before the feedback tone which terminated operant control on behavior after successful stopping (Fig. 1a). Variations in neural activity across SSDs were tested against a variety of parameters derived from different theories of MFC function described below (Fig. 2b; Supple-mentary Table 2).

Performance of the stop-signal task is explained as the outcome of a race between stochastic GO and STOP processes[26], instantiated by specific interactions enabling the interruption of the GO process by a STOP process[29,30] (Fig. 2a). A theory of medial frontal function posits

that it encodes the conflict between mutually incompatible processes[31]. Such conflict arises naturally as the mathematical product of the activation of the mutually incompatible GO and STOP units. The probability of noncanceled trials at each SSD, $p(NC|SSD)$, served as a proxy for conflict because it is the outcome of the race between the conflicting processes. This was validated by simulations of the GO and STOP units in the interactive race model (Fig. 2a). Therefore, the conflict model predicts variations in neural activity as a function of $p(NC|SSD)$.

Inspired by reinforcement learning models, we determined whe-ther neural modulation after stop-signal appearance varied with the likelihood of error associated with an experienced SSD[32]. Note that a stop-signal appearing after RT cannot contribute to this association. The experienced SSD can be learned only in trials when the stop-signal was seen ($SS_{seen}$), which are canceled trials and noncanceled trials that are explicit errors ($NC_{error}$). Therefore, the error-likelihood model predicts that neural activity varies with the likelihood of error, which was operationalized by $p(NC_{error}|SSD)/p(SS_{seen}|SSD)$. Neural modula-tion scaling positively (negatively) with this quantity encodes the error (success) likelihood associated with each SSD. This quantity diverges from $p(NC|SSD)$ at longer SSDs (Fig. 2b, top-left panel).

Finally, due to the temporal regularities of SSD and other events, we determined whether neural modulation signaled interval timing[14,15,33,34], moment-by-moment expectation for events[35,36], or the surprise associated with the violation of the expectations[37,38] (Fig. 2b, Supplementary Table 2, Supplementary Fig. 3). Guided by previous research on time perception[33,35,39], we tested whether neural modula-tion varied with elapsed time in linear or logarithmic scales. Expecta-tion was operationalized by hazard rate

$$h(t) = \frac{f(t)}{[1 - F(t)]} \tag{1}$$

where $f(t)$ is the probability density and $F(t)$ is the associated cumula-tive distribution. Surprise at the violation of this expectation was operationalized by Shannon's information,

$$s(t) = -\log_2[h(t)] \tag{2}$$

Different hazard rates and surprise quantities were computed based on different representations of temporal statistics plus the perceptual precision of SSD and the time until the feedback tone ($T_{tone}$; Fig. 2b; Supplementary Fig. 3a). Hazard rate models predict that neural activity before an event varies with $h(t)$, and surprise models predict that neural activity after an event varies with $s(t)$. The various models resulting from these quantities were compared through mixed-effects model comparison with Bayesian Information Criteria (BIC).

**Conflict monitoring neurons.** The classification pipeline identified 75 neurons with transient facilitation on canceled trials compared to latency-matched no-stop-signal trials (Fig. 1c). This facilitation was not a visual response to the stop-signal because it did not occur on non-canceled trials (Supplementary Figs. 1e and 2b). It also cannot con-tribute to reactive response inhibition because for nearly all neurons (71/75) it arose after SSRT (Fig. 3a, Supplementary Fig. 4a). The facil-itation started 90.4 ± 74 ms after SSRT with peak recruitment at ~110 ms at which time ~60% of neurons were active (Fig. 3a).

Through model comparison, we assessed how the magnitude of this modulation varied with task and performance parameters. The facilitation was best described by the conflict model with higher activity associated with larger $p(NC|SSD)$ (Fig. 3e; Supplementary Table 3, Mixed-effects linear regression (two-tailed) grouped by neu-ron, $t(212) = 4.24$, $p < 0.001$). Time-based and surprise models derived from the temporal statistics on canceled trials were candidates ($\Delta BIC < 2$). The error-likelihood ($\Delta BIC = 3.52$) and other surprise

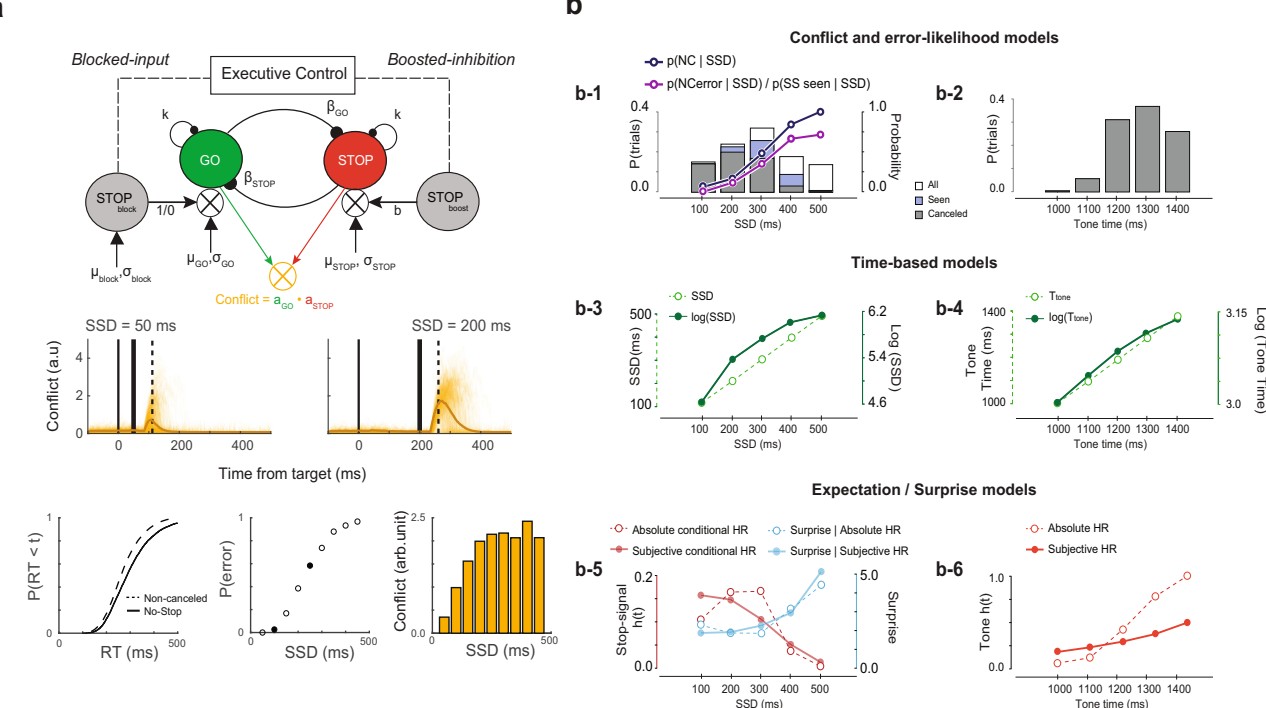

**Fig. 2 | Models of neural modulation. a** Interactive race model architecture. In interactive race architectures[29,30] GO and STOP units are racing, competing response channels, so the product of their co-activation measures response conflict in this task. In simulations with parameters simulating observed RT and $p$(NC|SSD) values (lower panels) the instantaneous product of $a_{GO}*a_{STOP}$ of individual (thin) and average (thick) increases around and peaks after SSRT (middle panels). The product of the activation of GO and STOP units producing countermanding performance is proportional to $p$(NC|SSD) (lower right panel). Consequently, $p$(NC|SSD) is an effective proxy for the conflict measure. **b** Quantities related to different behavioral and task parameters that arise from the variability in SSD and $T_{tone}$ are shown for a representative session. These quantities defined models with different predictions about variations in neural activity. Conflict and error-likelihood models (**b-1**) were based on the proportions of stop-signal trials (left ordinate) in which the saccade was successfully canceled (gray bars), those in which the stop-signal was seen because of RT > SSD (SS_seen; blue bars) and all stop-signal trials (white bars).

The probability of NC trials as a function of SSD (right ordinate) indexed conflict with $p$(NC|SSD) and error-likelihood with $p$(NC_error|SSD)/$p$(SS_seen|SSD), which diverge at long SSD. Time-based models were based on the duration of SSD (**b-1**) and $T_{tone}$ (**b-2**) with $t$ (left ordinate) and log($t$) (right ordinate) values (**b-3**, **b-4**). Expectation and surprise models were based on hazard rates of SSD and $T_{tone}$ derived from their respective distributions (**b-1** and **b-2**). Absolute and Subjective, incorporating imprecision in duration estimation, hazard rates (red) of SSD (**b-5**) and tone (**b-6**) and the surprise associated with stop-signal (**b-5**; blue) are shown. Other variants were also considered based on the underlying assumption about knowledge of task structure and imprecision in duration estimation (Supplementary Table 2, Supplementary Fig. 3a). Expectation of SS was quantified by hazard rate conditional on stop-signal trial probability (~40%). Surprise as a violation of expectations was quantified by the Shannon information derived from the hazard rate.

---

models earned weak support (ΔBIC = 5.7) or were rejected (ΔBIC > 6) (Fig. 3d; Supplementary Table 3). Although some surprise and time-based models explained the modulation, preference for the conflict model aligns with an earlier conjecture that these neurons signal conflict derived from the co-activation of GO and STOP unit activation[16]. Because this signal arises too late to influence response inhibition, we conjecture that it contributes to conflict monitoring.

On canceled trials, a minority of these neurons produced persistent weak activity until the tone, some with a transient response thereafter (Fig. 3a, b; Supplementary Fig. 1e). The spike rate immediately before the feedback tone was unrelated to its time or anticipation. Although RT slowed after successful stopping (multiple linear regression (two-tailed) controlling for direction and session, $t$(13664) = 15.7, $p$ < 0.001), the modulation of only a few neurons (7/75) covaried with RT (multiple linear regression controlling for SSD, $p$ < 0.05). Therefore, Conflict neurons were not involved in RT adjustments.

Over half of Conflict neurons (57%) exhibited multiplexing with reinforcement- or error-related signals reported previously[17] (Supplementary Table 7). Some produced higher discharge rates on unrewarded trials (previously identified as Loss neurons); some, had higher discharge rates on rewarded trials (Gain neurons). However, multiplexing incidence did not differ from that predicted by the sampling

prevalence of these signals (Chi-square test of homogeneity (one-tailed), $X^2$ (3, $N$ = 575) = 1.02, $p$ = 0.791). The vast majority (65/75) were not modulated when stopping failed (Supplementary Fig. 2b, Supplementary Table 7), reinforcing previous findings that conflict and error monitoring are distinct[16,17].

To investigate the microcircuit contribution of Conflict neurons, we examined their spike waveform duration and distribution across the layers (Supplementary Fig. 1a). Neurons were distinguished as broad- and narrow-spiking, which may identify pyramidal neurons that can send extrinsic connections and intrinsic inhibitory interneurons, respectively[40]. Most Conflict neurons (63/75) had broad spikes, but the incidence did not exceed that observed in the overall recording sample (Chi-square test of homogeneity (one-tailed), $X^2$ (1, $N$ = 575) = 0.205, $p$ = 0.152).

Conflict neurons were observed in all recording sites but more prevalently at some (Chi-square test of homogeneity (one-tailed), $X^2$ (4, $N$ = 575) = 11.6, $p$ = 0.020). Those recorded in sessions with perpendicular penetrations (36/75) revealed the spatiotemporal progression of the conflict signal across time and cortical depth (Fig. 3b). These neurons were sampled from all layers with an incidence corresponding to the sampling distribution (Chi-square test of homogeneity (one-tailed), $X^2$ (4, $N$ = 293) = 4.28, $p$ = 0.369; Fig. 3b; Supplementary Fig. 1a; Supplementary Table 1). The onset time of the modulation did not

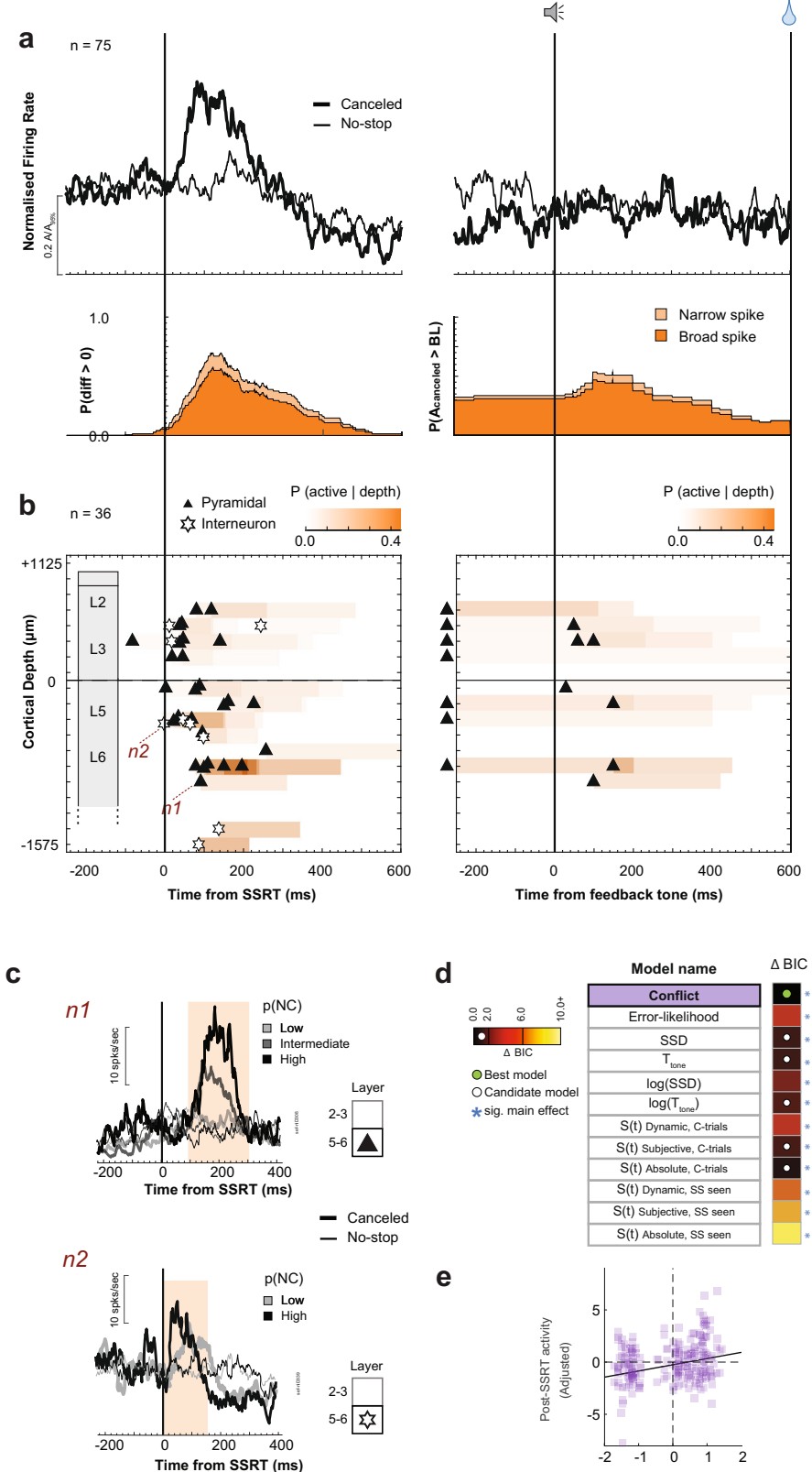

differ between L2/3 and L5/6 (Mann–Whitney $U$ test (two-tailed), $U(13, 23) = 181$, $z = -1.94$, $p = 0.052$). A larger proportion of concurrently activated Conflict neurons was observed in L5/6 (Fig. 3b). The few neurons modulating with the tone were observed in all layers.

Thus, complementing previous observations[16], particular neurons in SEF modulate in a manner consistent with signaling the co-activation of gaze-shifting (GO) and gaze-holding (STOP) mechanisms previously interpreted as conflict[31,41]. These neurons are distributed across all SEF layers and are predominantly broad-spiking, as expected from the sampling distribution. This conflict signal has also been observed in dopamine (DA) neurons in dorsolateral substantia nigra pars compacta (SNpc) during saccade countermanding[42]. The SNpc neurons

**Fig. 3 | Conflict neuron time–depth organization. a** Average spike rate (top) and recruitment (bottom) of broad- (dark) and narrow-spiking (light) neurons on canceled (thick) and latency-matched no-stop-signal (thin) trials, aligned on SSRT (left) and tone (right), normalized to 95th percentile within respective intervals. SSRT-aligned recruitment was the difference between trials indicated by $p$(diff > 0). Tone-aligned recruitment was the difference in spiking on canceled trials ($A_{canceled}$) relative to baseline (BL) lowest spiking value ±500 ms from the tone. Modulations after tone were not analyzed. Post-saccadic spiking on no-stop-signal trials before tone can exceed that on canceled trials. **b** Time–depth plot showing latency and recruitment across depth from perpendicular penetrations. Symbols mark the beginning of modulation for broad- (triangles) and narrow-spiking (stars) neurons. Color map indicates percentage through time at each depth relative to sampling density. Solid horizontal line marks L3–L5 boundary. The lower boundary of L6 is not discrete. **c** Modulation on canceled (thick) relative to latency-matched no-stop-signal (thin) trials for lower, intermediate, and higher $p$(NC|SSD) of two representative neurons: n1 had broad spikes in L6; n2 had narrow spikes in L5 (n2) (identified in **b**). Shaded interval highlights significant differences in spiking across conditions. **d** Model comparison table listing each tested model. The heatmap shows the difference in BIC values (ΔBIC) for each model compared to the model with the lowest BIC value (black fill) with hotter colors corresponding to lower ΔBIC values. Asterisks (*) indicate models with a significant main effect. The green circle indicates the best-fit model; the white indicates candidate models (ΔBIC < 2). Spike rate variation after SSRT was best predicted by the conflict model. Full statistics in Supplementary Table 3. **e** Significant variation of spiking (residualized and adjusted for spiking across neurons) as a function of $p$(NC| SSD) proxy for conflict (normalized $z$-scale). Each point plots average spike density and mean p(NC|SSD) across all trials in early-, mid-, or late-SSD bins for each of 75 neurons. For 11 neurons no reliable estimate of spike density for the late-SSD bin was obtained due to too few trials. These data points were not included.

modulate significantly earlier than those in SEF but accounting for the long conduction delay, a dopaminergic signal cannot cause the SEF modulation (Supplementary Fig. 9; Supplementary Table 9).

**Event timing neurons.** The classification pipeline identified another group of 84 neurons with ramping activity following target presentation on all trials with an abrupt reduction in discharge rate after SSRT on canceled trials (Fig. 4; Supplementary Fig. 1c–e). This modulation cannot contribute to reactive response inhibition because for nearly all neurons (76/84) it happened after SSRT (Supplementary Figs. 1d and 5a). This suppression occurred $62 \pm 58$ ms after SSRT, significantly earlier than Conflict neurons (Mann–Whitney $U$ test (two-tailed), $U$(84, 75) = 6001, $z = -2.44$, $p = 0.014$). By ~150 ms after SSRT, nearly all neurons had suppressed spiking (Fig. 4a). Noting the similarity of the ramping to earlier descriptions of time-keeping neurons[14,15], we focused on the ramping activity preceding the suppression.

Model comparison revealed that the ramping magnitude varied best with log(SSD) (Mixed-effects linear regression (two-tailed) grouped by neuron, $t$(250) = 12.62, $p = 0.001$) with higher activity for longer SSD durations (Fig. 4e; Supplementary Table 4). Absolute SSD earned weak support (ΔBIC = 2.7), but hazard rate models were rejected (ΔBIC > 6). Once successful stopping occurred, these neurons were suppressed. On no-stop-signal and noncanceled saccade trials, the ramping activity occurred in other epochs followed by a decline following the saccade or feedback tone (Supplementary Fig. 2b). Because the discharge rate decreased sharply after SSRT on canceled but not on noncanceled stop-signal trials (Fig. 4d, Supplementary Figs. 1e and 2b), we conjectured that these neurons are sensitive to the timing of events leading to successful stopping and not the timing of the stop-signal appearance per se.

Inherent to the task, following SSD and SSRT on canceled trials monkeys had to maintain fixation on the stop-signal for a variable but predictable duration ($T_{tone}$, Fig. 1a). Following the post-SSRT suppression, the activity in a subset of these neurons (38/84) exhibited a second ramping period preceding the tone, whereupon the spike rate decreased (Fig. 4; Supplementary Fig. 5b). This ramping had lower slope than that before SSRT. In a few neurons, the decrease in activity after the tone followed a brief transient response (Fig. 4a).

The variation in ramping dynamics before the tone was best explained by time-based models ($T_{tone}$; Mixed-effects linear regression (two-tailed) grouped by neuron, $t$(112) = 3.41, $p < 0.001$; Fig. 4h, Supplementary Table 4) with weak support for hazard rate models (3.0 < ΔBIC < 4.4). The $T_{tone}$ and log($T_{tone}$) models were indistinguishable (ΔBIC < 0.1) (Fig. 4g). The termination of this ramping activity was synchronized with the feedback tone and not the time at which fixation was interrupted following the feedback (Supplementary Fig. 5c).

Because the ramping activity of these neurons co-varies with SSD and $T_{tone}$ followed by suppression when the interval elapses, we conjecture that these neurons signal event timing. Event Timing neurons

were identified by ramping before SSRT, but ~45% also encoded the timing of the feedback tone. This suggests that the timing signal can exhibit different specificities in different neurons.

Next, we examined how the activity of these neurons relates to adjustments in behavior. Replicating previous findings, we found that RT was influenced by the SSD experienced in the previous trial, with slower RTs following longer SSD (Linear regression model (two-tailed), $t$(83) = 2.64, $p = 0.010$)[43]. However, the peak of the peri-SSRT ramping activity of only two neurons predicted RT in the trial following successful stopping (multiple linear regression (two-tailed) for each neuron controlling for SSD, $p < 0.05$). Therefore, this putative time-keeping signal does not influence the slowing of RT after canceled trials.

Neurons identified with event timing multiplexed with performance monitoring signals reported previously[17]. Event timing was significantly more likely to be observed in Error and Gain neurons compared to Loss neurons (Chi-square test of homogeneity (one-tailed), $X^2$ (3, $N = 575$) = 44.86, $p < 0.001$, Supplementary Table 7). The nature of the ramping before SSRT was invariant across multiplexing associations. Ramping after the feedback tone until reward delivery was observed in Gain neurons (24/84).

To elucidate the microcircuit contribution of Event Timing neurons, we examined their spike waveform duration and distribution across the layers. The majority (72/84) were broad-spiking similar in proportion to the sampling distribution (Chi-square test of homogeneity (one-tailed), $X^2$ (1, $N = 575$) = 3.75, $p = 0.053$). These neurons were found in all penetrations but were more common in some sites ($X^2$ (4, $N = 575$) > 39.3, $p < 0.001$; Supplementary Table 1). The time–depth organization of these neurons was revealed in the sample from perpendicular penetrations (49/84; Fig. 4b). Those ramping before SSRT were found across all layers in proportion to the sampling distribution (Chi-square test of homogeneity (one-tailed), $X^2$ (4, $N = 293$) = 7.33, $p = 0.120$; Fig. 4b; Supplementary Fig. 1a; Supplementary Table 1). However, those with ramping after SSRT until the tone were significantly more concentrated in lower L3 and L5 ($X^2$ (2, $N = 293$) = 10.37, $p = 0.006$; Supplementary Table 1). The suppression time after SSRT or the tone did not vary across layers.

Thus, neurons in SEF exhibited ramping activity that can signal the time preceding critical events for successful task performance. These results show that these neurons are distributed across all SEF layers and are predominantly broad-spiking as expected from the sampling distribution. Also, the time-related signals in the SEF can have different functional specificities[44] and multiplex with error and reinforcement signals in different layers.

**Goal maintenance neurons.** We identified another class of 54 facilitated neurons with significantly greater discharge rate on canceled compared to latency-matched no-stop-signal trials after SSRT (Fig. 5). This neuron class was distinguished from Conflict neurons based on

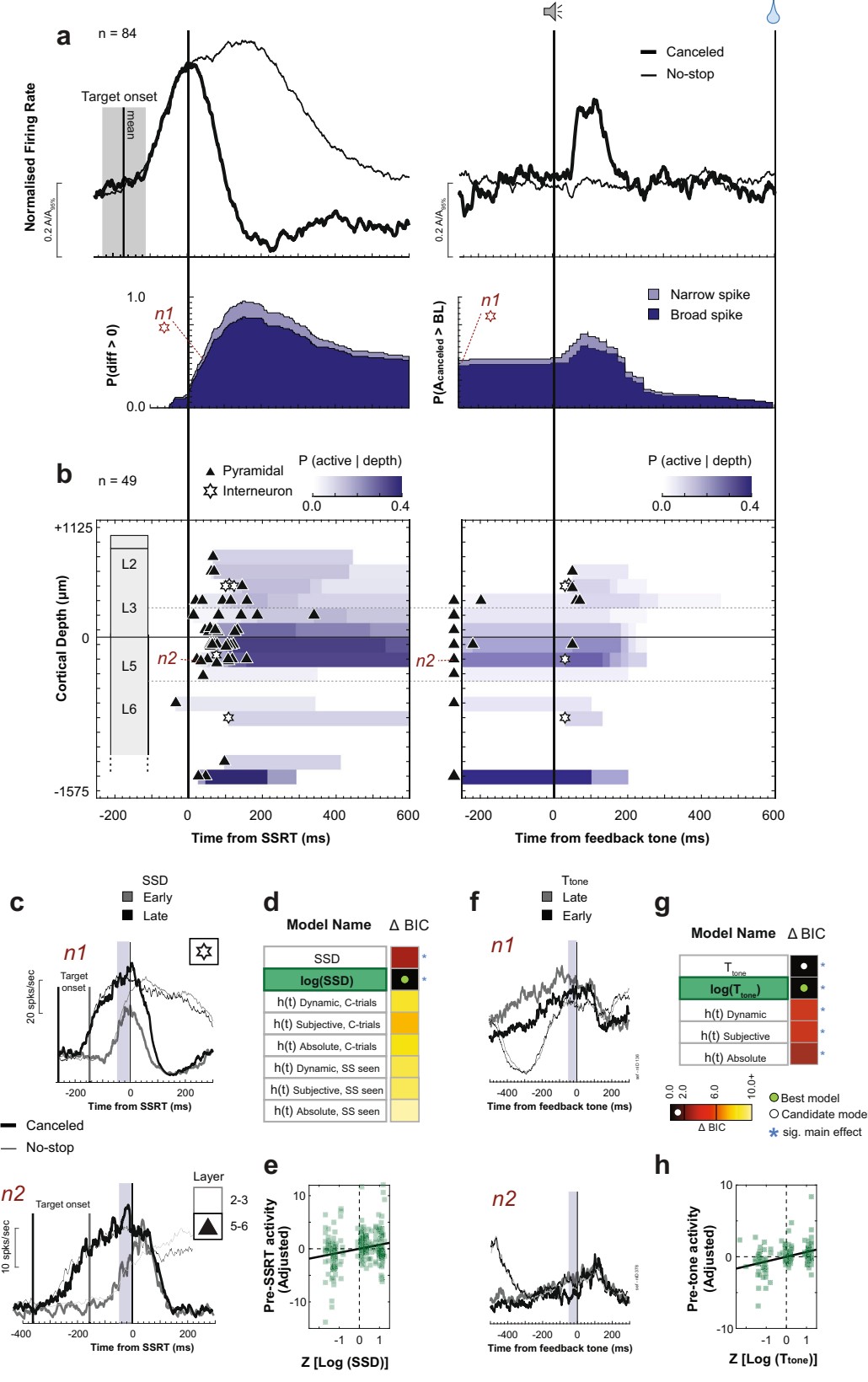

the duration of facilitation (*K*-means clustering, Fig. 1c, Supplementary Fig. 1d) and by other differences described below.

The facilitation was not a response to the stop-signal, because it did not occur on noncanceled trials (Supplementary Fig. 2b). It cannot contribute to reactive response inhibition because it arose after SSRT for effectively all neurons (53/54; Fig. 5a; Supplementary

Fig. 6a). On average, the facilitation began $120 \pm 85$ ms after SSRT not significantly different from that of Conflict neurons (Mann–Whitney $U$ test (two-tailed), $U(75, 54) = 4509$, $z = -1.75$, $p = 0.081$) but significantly later than the suppression in Event Timing neurons (Mann–Whitney $U$ test (two-tailed), $U(84, 54) = 4940.5$, $z = -3.91$, $p < 0.001$). The peak recruitment of these

**Fig. 4 | Event timing neuron time–depth organization.** Conventions as in Fig. 3. **a** Average spike rate and suppression time of neurons with ramping spiking after the target (mean: vertical line; min-max: gray-shaded rectangle) followed by suppression on successfully canceled relative to latency-matched no-stop-signal trials. SSRT-aligned recruitment was the difference in spiking between canceled and no-stop-signal trials. **b** Time–depth plots. Horizontal dashed lines highlight where Event Timing neurons with pre-tone ramping were concentrated in lower L3 and L5. **c** Modulation on successfully canceled relative to latency-matched no-stop-signal trials for early (lighter) and late (darker) SSDs of two representative neurons: n1 had narrow spikes but no layer assignment; n2 had broad spikes in L5. Spiking in shaded 50 ms before SSRT (shaded) was used for analysis. **d** Model comparison table for pre-SSRT activity. The best model is highlighted in green. Variation of spiking was best predicted by log(SSD). Full statistics in Supplementary Table 4. **e** Significant variation of spiking activity before SSRT as a function of log(SSD) with 84 neurons contributing 252 samples. **f** Modulation of neurons n1 and n2 with ramping before the tone. Shaded 50 ms indicates epoch analyzed. **g** Model comparison table for pre-tone activity. Variation of spiking was best-predicted log($T_{tone}$). Full statistics in Supplementary Table 4. **h** Significant variation of spiking activity before tone as a function of log($T_{tone}$) with 38 neurons contributing 144 samples across early, intermediate, or late $T_{tone}$.

---

neurons reached ~95% after ~300 ms, later than that of the Conflict (~110 ms) and Event Timing neurons (~150 ms) (Fig. 5a).

The variation in activity of these neurons was explained best by the surprise model (S(t)$_{Subjective, SS\ seen}$; Mixed-effects linear regression (two-tailed) grouped by neuron, $t(151) = -3.91$, $p < 0.001$; Fig. 5d; Supplementary Table 5) with other surprise and time-based models also candidates ($\Delta BIC < 2$) and weak support for log(SSD) ($\Delta BIC = 3.0$) and conflict ($\Delta BIC = 4.1$) models. The error-likelihood model was rejected ($\Delta BIC > 6.0$). The variation of activity varied inversely with surprise and positively with $T_{tone}$ and log($T_{tone}$) (Fig. 5c, e; Supplementary Table 5). Thus, the modulation of these neurons was further distinguished from the conflict signal by the different (and opposite) relationship to performance and task parameters (Supplementary Fig. 6e).

On canceled trials after SSRT a large fraction of these neurons (40/54) produced persistent spiking; a fraction exceeding Conflict neurons (Chi-square test of homogeneity (one-tailed), $X^2$ (1, $N = 129$) = 27.3, $p < 0.001$). The sustained activity attenuated ~300 ms after the feedback tone that cued successful performance (Fig. 5a). Attenuation after the tone was also observed on no-stop-signal trials. The spike rate immediately before the feedback tone was unrelated to any factor related to its time or anticipation. Consistent with the indirect contribution of SEF to saccade initiation, the termination of this modulation was time-locked to the tone and not when monkeys stopped fixating on the stop-signal (on canceled trials) or the target (on no-stop-signal trials). Hence, this signal is not directly involved in gaze-holding (Supplementary Fig. 6c).

Based on the evidence above and previous findings identifying SEF signals with working memory[11,12], we conjecture that these neurons contribute to maintaining a representation of task goals (e.g., sustain unblinking fixation) for the successful completion of the task. Consistent with this hypothesis, when the monkey broke fixation too early, the facilitation after SSRT was reduced significantly in a subset of neurons with enough data (14/54; Supplementary Fig. 6f). Therefore, we refer to these neurons as Goal Maintenance neurons.

The modulation of only a minority (7/54) of these neurons covaried with RT on the subsequent trial (multiple linear regression (two-tailed) controlling for SSD, $p < 0.05$) which precludes this signal from contributing to adjustments of RT.

Goal Maintenance neurons multiplexed with reinforcement and error signals[17]. The vast majority were previously classified as Loss neurons because although the activity of most of these neurons was suppressed after the feedback tone cued success when it cued failure, activity increased[17] (Supplementary Fig. 6d). The prevalence of this multiplexing pattern exceeded chance (Chi-square test of homogeneity (one-tailed), $X^2$ (3, $N = 575$) = 19.43, $p < 0.001$; Supplementary Table 7).

To elucidate the microcircuit contribution of Goal Maintenance neurons, we examined spike duration and distribution across the layers. Over one-third of Goal Maintenance neurons were narrow-spiking, a proportion exceeding chance sampling (21/54; Chi-square test of homogeneity (one-tailed), $X^2$ (1, $N = 575$) = 9.27, $p = 0.002$). They were found in all penetrations but significantly more commonly at certain sites ($X^2$ (4, $N = 575$) > 39.3, $p < 0.001$, Supplementary Table 1). Perpendicular penetrations revealed the time-depth organization of 34

Goal Maintenance neurons (Fig. 5b). The distribution of these neurons across cortical layers was significantly different from the sampling distribution ($X^2$ (4, $N = 293$) = 11.24, $p = 0.024$, Supplementary Fig. 1a, Supplementary Table 1) with significantly more in L2/3 relative to L5/6 (Fig. 5b, $X^2$ (1, $N = 293$) = 10.37, $p = 0.001$). Their laminar distribution was also significantly different from those of Conflict ($X^2$ (1, $N = 70$) = 11.54, $p < 0.001$) and Event Timing neurons ($X^2$ (1, $N = 83$) = 5.49, $p = 0.019$). Those in L2/3 modulated significantly earlier than those in L5/6 (L2/3: $85 \pm 64$ ms, L5/6: $193 \pm 101$; Mann–Whitney $U$ test (two-tailed), $U(26,8) = 388$, $z = -2.7$, $p = 0.007$).

Thus, consistent with previous studies[11,12], neurons in SEF produced activity sufficient to enable a working memory representation of the goal of saccade inhibition through time tuned by experienced intervals. These results show that these neurons are most common in L2/3 and a relatively higher proportion have narrow spikes.

**Functional classification of N2/P3 ERP related to response inhibition.** To determine whether macaque monkeys produce ERP components associated with response inhibition homologous to humans[7], we sampled EEG from an electrode located over MFC (10–20 Fz) while recording neural spiking in SEF (Fig. 6a). To isolate signals associated with response inhibition by eliminating components associated with visual responses and motor preparation, we measured the difference in polarization on canceled trials and latency-matched no-stop-signal trials for each SSD (Fig. 6b). Homologous to humans, we observed an enhanced N2/P3 sequence associated with successful stopping. The conclusions drawn from the results presented below do not differ if the analyses are performed on the raw EEG polarization on canceled trials instead of the difference between conditions in these intervals.

The N2, characterized as a negative deflection homologous to the human N2, began ~150 ms and peaked at $222 \pm 17$ ms after the stop-signal, well after the visual ERP polarization (Supplementary Fig. 7a). The N2 was observed after SSRT, too late to index reactive response inhibition. Furthermore, the N2 peak time across sessions was significantly better aligned on stop-signal presentation than on SSRT, further dissociating the N2 from reactive inhibition (F-test comparison of variances (two-tailed), $F(28,28) = 0.29$, $p = 0.002$; Supplementary Fig. 7c). Variation in the amplitude of the N2 was only explained by the error-likelihood model with the largest negativity associated with the lowest error-likelihood (mixed-effects linear regression (two-tailed) grouped by session, $t(85) = 2.42$, $p = 0.018$; Fig. 6d, Supplementary Table 6). Conflict, time-based, and surprise models were rejected (non-significant main effect, and $\Delta BIC > 3.0$; Fig. 6c). This result adds to the inconclusive evidence for the frontal N2 association with conflict monitoring and response inhibition[7].

The N2 was followed by a robust P3 beginning ~300 ms and peaking $358 \pm 17$ ms after the stop-signal, homologous to the human P3[7] (Fig. 6a, b). The peak polarization time was better synchronized on the stop-signal than on SSRT (F-test comparison of variances (two-tailed), $F(28,28) = 0.44$, $p = 0.034$; Supplementary Fig. 7c). Variation in the amplitude of the P3 was best described by the log($T_{tone}$) on canceled trials, with P3 polarization increasing with $T_{tone}$ (Mixed-effects linear regression (two-tailed) grouped by session, $t(85) = 3.72$, $p < 0.001$; Fig. 6c, e; Supplementary Table 6). All time-based models

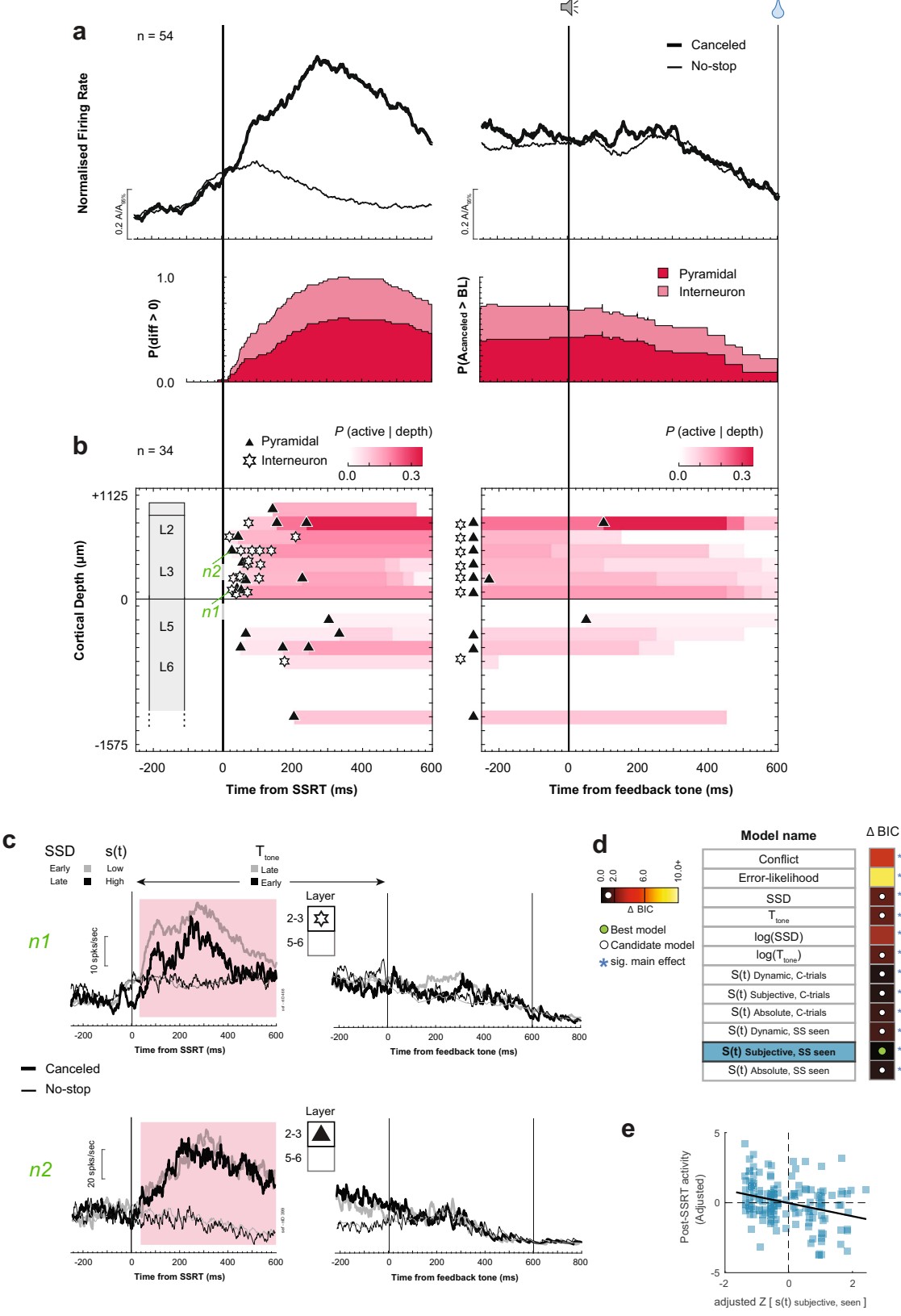

were candidates (ΔBIC < 1.30). The conflict (ΔBIC = 2.90) and most surprise models (4.5 < ΔBIC < 5.7) received weak support, and error-likelihood and one surprise model were rejected (ΔBIC > 6.0).

**Association of N2/P3 with neural spiking.** We examined how neural spiking related to concomitant ERP[17,45]. Appreciating that EEG arises

from ~10⁶ neurons and spikes are too brief to create scalp EEG[46], we evaluated whether the N2/P3 complex can be a biomarker of layer-specific neural spiking.

The N2 coincided generally with the peak recruitment of Conflict and of Event Timing neurons, and the P3 with the peak recruitment of Goal Maintenance neurons (Fig. 6b). The relationship between neural

**Fig. 5 | Goal maintenance neuron time–depth organization.** Conventions as in Fig. 3. **a** Average spike rate and recruitment through time of neurons with persistent activity on canceled relative to latency-matched no-stop-signal trials. **b** Time–depth plot. **c** Modulation on successfully canceled relative to latency-matched no-stop-signal trials for shorter and longer $T_{tone}$ of two representative neurons: n1 had narrow spikes in L3; n2 had broad spikes L3. Spiking of both neurons decreased after the tone. Shaded interval highlights significant differences in spiking across

conditions. **d** Model comparison table for post-SSRT activity. Variation of spiking was best predicted by the surprise model $S(t)_{Subjective, SS\ seen}$. Full statistics are in Supplementary Table 5. **e** Significant variation of spiking activity after SSRT as a function of $S(t)_{Subjective, SS\ seen}$ with 54 neurons contributing 162 samples across early, intermediate, or late $T_{tone}$. For 9 neurons no reliable estimate of spike density for the late-SSD bin was obtained due to too few trials. These data points were not included.

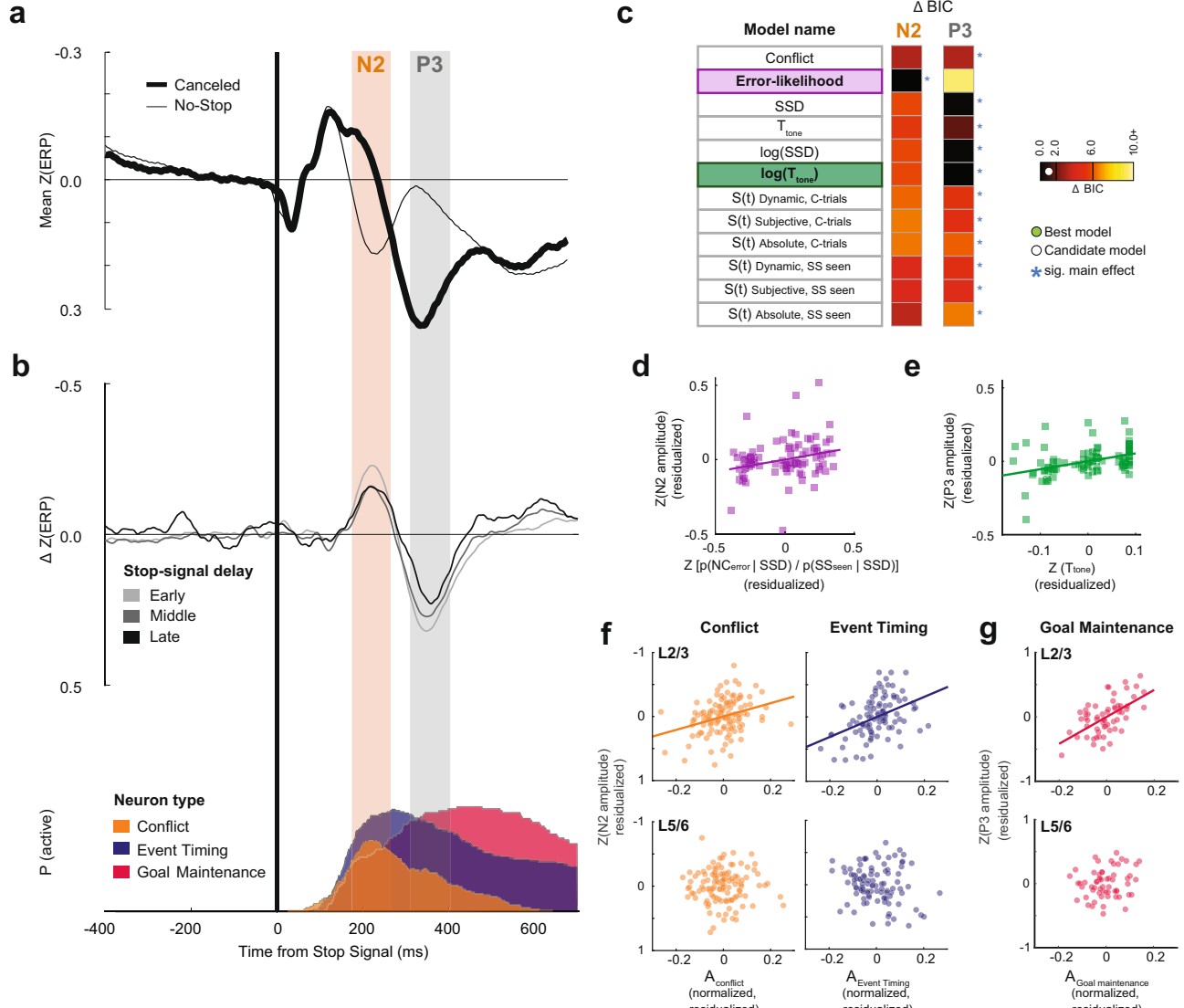

**Fig. 6 | Event-related potentials for successful response inhibition.** Conventions as in Fig. 3. **a** Grand average *z*-transformed EEG on canceled (thick) and latency-matched no stop-signal (thin) trials. **b** Difference functions (top) remove stimulus-evoked ERP to highlight N2 and P3 components in 3 SSD bins. Shaded intervals show ±50 ms sampling interval around N2 (orange) and P3 (gray) peaks. Concomitant recruitment of the three neuron classes (bottom). **c** Model comparison shows N2 amplitude variation was best described by the error-likelihood, and P3 amplitude was best described by log($T_{tone}$). Full statistics in Supplementary Table 6. **d** Significant variation of N2 amplitude as a function of $p(NC_{error}|SSD)/p(SS_{seen}|SSD)$ in 29 sessions contributing 87 points across early, intermediate, or late SSD. **e** Significant variation of P3 amplitude as a function of $T_{tone}$ in 29 sessions contributing 87 points across $T_{tone}$. **f** Partial regression between N2 amplitude and

spike rate for Conflict (left) and Event Timing (right) neurons in L2/3 (top) and L5/6 (bottom) for sessions with both L2/3 and L5/6 neurons sampled. Ordinate scale plots, with EEG convention, residual from fixed-effects-adjusted ERP amplitude controlling for activity in the opposite layer. Abscissa scale plots residual fixed-effects-adjusted neuronal discharge rate in the identified layer controlling for the activity in the opposite layer and stop-signal delay. Each point plots the average EEG voltage and associated spiking rate in one of 20 bins with equal numbers of trials per session. Plotted are 120 points from 6 sessions for Conflict (left) and 100 points from 5 sessions for Event Timing (right) neurons. N2 amplitude variation was predicted by spiking rate variation of Conflict and Event Timing neurons in L2/3 but not in L5/6. **g** P3 amplitude variation was predicted by spiking rate variation of Goal Maintenance neurons in L2/3 but not in L5/6.

events in SEF and cranial voltages is both biophysical and statistical. The cranial voltage produced by synaptic currents associated with a given spike must follow Maxwell's equations applied to the brain and head, regardless of the timing of the different events. Hence, we

counted the spikes of the three classes of neurons separately in L2/3 and in L5/6 during the 100 ms spanning the peak of the ERP and tested multiple linear regression models with activity in upper layers (L2/3) and lower layers (L5/6) of each neuron class as predictors. Only

successfully canceled trials were included in this analysis. We found that variation in N2 voltage is not associated with the spiking of Goal Maintenance neurons (Multiple linear regression (two-tailed) with L2/3 and L5/6 activity as predictors; L2/3: $t(57) = -1.28$, $p = 0.206$; L5/6: $t(57) = 0.52$, $p = 0.605$; Supplementary Fig. 8a; Supplementary Table 8). However, it was predicted by the spiking in L2/3 but not in L5/6 of Conflict (L2/3: $t(117) = -3.6$, $p < 0.001$; L5/6: $t(117) = 0.046$, $p = 0.963$) and of Event Timing (L2/3: $t(97) = -4.60$, $p < 0.001$; L5/6: $t(97) = 1.67$, $p = 0.097$) neurons (Fig. 6f). When the discharge rate of the L2/3 neurons was higher, the N2 exhibited more negativity. N2 polarization was also predicted by the spiking in L2/3 but not in L5/6 of other neurons that were not modulated on canceled trials and so were not described in this manuscript (L2/3: $t(317) = -2.51$, $p = 0.012$; L5/6: $t(317) = -1.60$, $p = 0.110$; Supplementary Fig. 8a). Conversely, variation in P3 polarization was predicted by the spiking activity of Goal Maintenance neurons in L2/3 but not L5/6 (multiple linear regression (two-tailed) with L2/3 and L5/6 activity as predictors (L2/3: $t(57) = 5.46$, $p < 0.001$; L5/6: $t(57) = 1.47$, $p = 0.148$; Fig. 6g; Supplementary Fig. 8c, d; Supplementary Table 8), with higher spike rates associated with greater P3 positivity. P3 amplitude was unrelated to the spiking of Conflict (L2/3: $t(117) = 0.44$, $p = 0.660$; L5/6: $t(117) = -0.49$, $p = 0.624$), Event Timing (L2/3: $t(97) = -1.19$, $p = 0.236$; L5/6: $t(97) = -0.77$, $p = 0.440$), or unmodulated neurons (L2/3: $t(317) = -1.11$, $p = 0.269$; L5/6: $t(317) = 0.054$, $p = 0.956$; Supplementary Fig. 8c; Supplementary Table 8).

## Discussion

These results offer further insights into the cortical microcircuitry supporting executive control in primates. Model-based analysis of the latency, temporal dynamics, and variation in the strength of neural spiking across the neuron sample revealed functionally distinct and theoretically informative classes of neurons with distinct biophysical and laminar properties. Moreover, a bridge between these neurophysiological findings and human electrophysiology was established through the specific associations observed between the N2 and P3 ERP observed in response inhibition tasks and classes of neurons in particular cortical layers.

The utility of these findings is amplified by their complementarity with our previous description of the laminar organization of error and reward processing in SEF[17]. Based on the results presented in this paper, we will discuss how SEF can contribute to conflict monitoring, time-keeping, and goal maintenance. Coupled with the current knowledge about the connectivity of SEF[47–49], our findings detailing the laminar distribution of neurons signaling response conflict, event timing, and maintaining goals suggest several specific hypotheses and research questions about how SEF and associated structures accomplish response inhibition and executive control (Fig. 7). Also, complementing our earlier description of the source of the ERN[17], we now report a macaque homolog of the N2/P3 ERP components associated with response inhibition. These results demonstrate one potential cortical source of these ERP components.

### Conflict, surprise, salience, and dopamine

In this study, we report a population of SEF neurons with pronounced, transient facilitation after successful response inhibition (SSRT). These neurons were predominantly broad-spiking, proportional to the sampling distribution, and found in all layers. Their spike rate during canceled trials was best described by the conflict model, operationalized by the probability of generating noncanceled saccades— $p(NC|SSD)$. The mechanism producing responses in this task is a well-understood network of gaze-shifting and gaze-holding neurons in the frontal eye field (FEF) and superior colliculus (SC)[27,50]. Neurocomputational models demonstrate how this network can instantiate the GO and STOP processes[29,30,51]. Noncanceled saccades happen when the gaze-holding STOP units do not interrupt the rise to the threshold of the gaze-shifting GO units. Reactive inhibition happens only if the

STOP unit interrupts the GO unit, which must be brief and potent. Thus, in successful canceled trials, the GO and STOP units (corresponding to gaze-shifting and gaze-holding neurons) are in an unstable state of co-activation, corresponding to the original, formal definition of conflict[31]. The multiplicative conflict between GO and STOP accumulator units scales with $p(NC|SSD)$ and peaks following SSRT (Fig. 2a) but it is unrelated to adjustments in RT. Therefore, we conjecture that this modulation signals the difficulty of the stopping process, which can then be incorporated with other information to drive adaptive changes in behavior.

Recent findings from the nigrostriatal dopamine system of monkeys performing saccade countermanding[42] offer an alternative interpretation for these neurons. The modulation of DA neurons in dorsolateral SNpc scales with $p(NC|SSD)$ just like the Conflict neurons. Although alternative models were not tested in that study, the spiking of DA neurons has been identified with salience or surprise[42,52]. We tested the surprise hypothesis by quantifying the moment-by-moment expectancy of the stop-signal given the experienced distribution of SSD and probability of stop-signal occurrence[37,38]. This neural modulation was explained almost as well by surprise as by conflict. However, it could not be explained by error-likelihood. While these SEF data align with the Conflict hypothesis, they do not exclude surprise or salience hypotheses. From the perspective of reinforcement theory, a phasic DA signal can be an eligibility trace broadcast to SEF and other regions to associate reinforcement with successful cancelation after the infrequent stop-signal. To be most useful, such an eligibility trace must be salient and may be surprising.

### Event timing and goal maintenance

We found neurons encoding the timing of task events. In our version of the stop-signal task, knowledge of the timing of the stop-signal and of the feedback tone was important. To earn the reward, monkeys must hold their gaze stable for an extended period, which required preventing eye movements and blinks that would interrupt the camera-based eye tracker. This entails learning and exploiting regularities in the timing of task events[43,53]. A contribution of SEF and nearby areas in action timing and time production tasks has been demonstrated[14,15,54]. We extend that description to this stop-signal task in terms of time keeping and goal maintenance.

A distinct group of SEF neurons produced ramping spike rates. When the saccade was inhibited, this ramping was interrupted by a pronounced suppression. These neurons were described previously with no explanation[8]. Our results rule out the possibility that these neurons control movement initiation because the suppression occurred too late to contribute to gaze-holding. Also, this ramping activity did not encode the expectancy for the stop-signal arising from the temporal distribution of SSDs and the probability of stop-signal appearance. Instead, they were best described by SSD.

The task design exposed a second period of ramping before the feedback tone in roughly half of these neurons which reached higher levels for longer durations. Our discovery of an association between the spiking rate and the log-transformed duration of the elapsed time motivates a more integrated interpretation framed by a body of research on time keeping[14,15,33,34,36,44,54]. We interpret the ramping activity as representing the timing of task events, like neurons in the basal forebrain that signal event timing depending on surprise, salience, and uncertainty[44]. The sharp suppression in activity can reset the system to track the timing of subsequent events. Although the stop-signal occurred randomly and response inhibition was accomplished stochastically, the feedback tone was certain to happen. Therefore, we conjecture that neurons exhibiting ramping activity before both SSRT and the feedback tone encode the timing of expected salient events regardless of certainty or expected response. In contrast, neurons with no ramping activity before the tone can encode events that are less certain in occurrence or consequence. These differences were

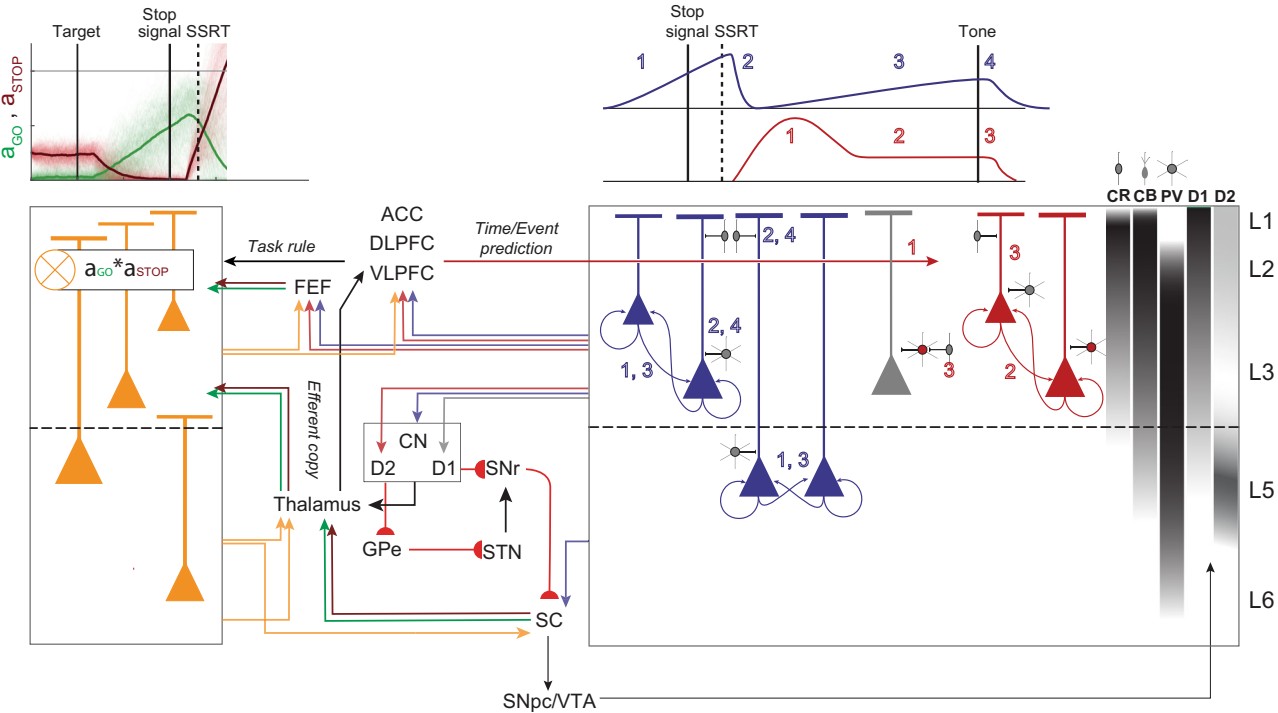

**Fig. 7 | Executive control circuitry.** Conflict (orange), Event Timing (dark blue), and Goal Maintenance (dark red) neurons are illustrated with selected anatomical connections and laminar densities of calretinin (CR), calbindin (CB), and parvalbumin (PV) neurons[21] and of D1 and D2 receptors[72] (right). Dopamine projections from SNpc and VTA modulate all computations in SEF. *Left,* Conflict signal can arise through coincident inputs of gaze-holding (STOP, dark red) and gaze-shifting neurons (GO, green) directly from FEF and indirectly via thalamus from SC terminating in L2 and L3. Conflict signals integrated across apical and basal dendrites can be sent to multiple cortical and subcortical structures. *Right,* Schematic profiles of Event Timing and Goal Maintenance activity with numbered phases. Event Timing neurons can receive inputs from DLPFC and ACC informing them about an upcoming event. Ramping results from recurrent connections (1, dark blue). SEF can receive information about the stop-signal appearance and successful stopping from VLPFC and DLPFC and from Conflict neurons. This signal can suppress the ramping activity via inhibitory connections onto Event Timing neurons (2, dark blue), resetting these neurons for the next ramping phase (3, dark blue), which is terminated by the appearance of the feedback tone (4). Event Timing neurons can project to the caudate nucleus (CN) to inform the fronto-striatal reinforcement learning loop about experienced event timing. Goal Maintenance neurons can delay unwanted movement through the push-pull basal ganglia circuitry. Pyramidal neurons can project to the indirect (D2) pathway and inhibitory neurons can project to other pyramidal neurons, unobserved in this study (gray), that project to the direct (D1) pathway. Inputs from DLPFC and ACC terminating in L2/3 can inform SEF of the anticipated timing of task events for successful completion of the task based on the experienced SSD. These inputs can produce the phasic response in Goal Maintenance neurons (1, red) followed by persisting activity via recurrent connections with balanced excitation and inhibition (2, red). The feedback tone, integrated with the task rule from DLPFC, terminates operant control on behavior through CR and CB inhibition of the sustained activity (3). Further details in the text.

reinforced by the distinct laminar distribution of the two groups of neurons.

Another class of neurons produced persistent activity on canceled trials after SSRT, for most neurons lasting until the feedback tone. These neurons were not gaze-holding neurons contributing to response inhibition because this facilitation occurred too late to be involved in reactive inhibition. In neurons with enough trials, we observed weaker modulation when monkeys aborted trials after canceling the saccade. Similar signals have been observed in tasks with a cue indicating reinforcement probability[55,56]. However, the activity of these SEF neurons was clearly not explained by error-likelihood, which is the inverse of success probability. Instead, time-based and surprise models could explain this modulation. The inverse relationship between spike rate and surprise implies the contribution of inhibitory neurons with spiking rates directly proportional to surprise. We did not sample many such neurons, but narrow-spiking Conflict neurons can serve this role. The direct relationship between spike rate and $T_{tone}$, on the other hand, resembles a common motif for encoding duration[34]. Furthermore, this activity was linked to events occurring after but not before successful stopping. Therefore, we believe that this modulation is explained most parsimoniously by $T_{tone}$ predicted by the experienced SSD and not surprise. To earn a reward on canceled trials, monkeys needed to sustain unblinking fixation on the stop-

signal until the feedback tone (Fig. 1a). Hence, we conjecture that these neurons contribute to goal maintenance. This conjecture is consistent with an original theory of response inhibition[26] and previous evidence linking SEF to working memory[11,12] and working memory to time estimation[57,58]. Future work can discriminate time-based and surprise parameters and evaluate the link between surprise and goal maintenance.

## Cortical microcircuitry of executive control

By combining the laminar distribution of the neuron classes described in this study with the anatomical, histological, and neurophysiological properties of SEF neurons, we offer hypotheses about mechanisms by which such signals can be generated and influence other neurons, layers, or brain areas (Fig. 7).

To signal conflict, SEF can be informed about the dynamic state of gaze-shifting and gaze-holding through inputs from FEF and oculomotor thalamic nuclei. Based on previous conjectures[5] and recent biophysical modeling[59] we hypothesize that the integration of information producing the modulation of these neurons is derived through synaptic integration across apical and basal dendrites. The circuitry sufficient for signaling prediction errors[19] can signal the occurrence of conflict in this task. The presence of this signal in all layers enables it to interact with all intrinsic processes and possibly influence all cortical

and subcortical efferent targets. For example, a thalamic input that a saccade has been canceled can change the corollary discharge communicated through the thalamus[60]. Consistent with its original conception, communicating conflict (or salience or surprise or just difficulty) to multiple areas simultaneously can coordinate adaptive changes in behavior[61].

The unexpected parallels between SEF and SNpc modulation patterns invite consideration of cause and effect. SEF is innervated by DA neurons in SNpc[62]. Whilst SNpc DA neurons modulated significantly earlier than the SEF Conflict neurons (Supplementary Fig. 9), the estimated arrival times of DA spikes in SEF were not significantly different from the modulation times of the Conflict neurons after accounting for the very slow conduction of DA axons[63] (~100 ms conduction time from SNpc to SEF, Supplementary Fig. 9) and second-messenger delay. Therefore, we infer that this transient modulation in SEF cannot be caused by DA inputs. Conversely, because axon terminals from SEF are rare in SNpc[47,49], SEF neurons are unlikely to directly cause the modulation of the SNpc DA neurons. Instead, other investigators have shown that the phasic DA activation is delivered by the SC[64]. Through the Conflict neurons in L5, SEF can influence SC directly[47]. Curiously, though, the modulation specifically after SSRT scaling with related performance parameters has not been observed in SC[50].

While Event Timing neurons were found in all layers, those encoding timing regardless of event predictability or action were most common in L3 and L5 with broad spikes consistent with pyramidal neurons. This laminar differentiation demonstrates that the timing of different events can be conveyed by different layer-specific extrinsic connections. The timing signal can be sent via cortico-cortical connections to other cortical areas to influence motor, cognitive, and limbic processes. Also, these neurons can contribute to fronto-striatal pathways to learn and update a representation of the temporal structure of the task[54,65,66]. Axon terminals from SEF are dense in the caudate nucleus[48], arising from pyramidal neurons in L3 and L5[67,68]. In fact, neurons with this pattern of modulation have been described in a recent investigation of the caudate nucleus of monkeys performing saccade countermanding[42]. Our finding that the suppression in the SEF Event Timing neurons occurred after SSRT, but significantly earlier than those previously reported in the caudate nucleus suggests a primary role of the cortex in this signaling (Supplementary Fig. 9)[42].

The rapid suppression of the ramping activity after SSRT merits consideration. One source can be intracortical inhibition from the narrow-spiking, putative parvalbumin (PV) neurons that we observed. Another source can be the very small calbindin (CB) and calretinin (CR) neurons concentrated in L2/3 that are innervated by the dorsolateral prefrontal cortex (DLPFC) and selectively inhibit pyramidal neurons[69], although our methods are unlikely to sample their spikes. We note that although SEF is an agranular structure with weak interlaminar inhibitory connections[22], CR neurons in L2/3 can potently inhibit L5 neurons through specialized projections on the apical dendrites[70]. This inhibition must be informed about the presence of the stop-signal and inhibition of the saccade. We observe that such a signal is available in the Conflict neurons. However, the suppression of Event Timing neurons occurred significantly earlier than the facilitation of the Conflict neurons. Further research can resolve these cortical interactions.

Goal Maintenance neurons were mainly found in L2/3. Inputs to these layers from DLPFC, ventrolateral prefrontal cortex (VLPFC), and anterior cingulate cortex (ACC) can signal task rules and the expected time of the secondary reinforcer when executive control can be released. Input from these areas can inform SEF of the anticipated reward based on the experienced stop-signal delay, contingent on successful stopping. Dopaminergic release in SEF from SNpc and ventral tegmental area (VTA), with similar time-predicting signals[71], can enhance these influences through the higher density of D1 relative to D2 receptors in L2/3[72]. The sustained discharge can be maintained

through recurrent activation within SEF and between other structures[12]. Also, many Goal Maintenance neurons had narrow spikes, consistent with PV inhibitory neurons, which can balance excitation and inhibition necessary for the maintenance of persistent activity in recurrent networks[73,74]. The auditory feedback tone, integrated with the task rule from DLPFC, cues the termination of operant control on behavior, resulting in the inhibition of pyramidal and inter-neurons by CR and CB neurons. This results in the termination of the sustained activity.

Pyramidal Goal Maintenance neurons can encourage the suppression of movements through projections to the indirect pathway D2 neurons in the striatum[67,68]. Intrinsic inhibitory Goal Maintenance neurons can suppress the movement by inhibiting pyramidal neurons projecting the direct pathway D1 and to the frontal eye field. As PV neurons in primates largely lack extrinsic connections[75], we propose that this can be mediated by the inhibition of other excitatory neurons (unidentified neurons and possibly Gain neurons identified in ref. 17) that send projections to these motor structures (gray neurons in Fig. 7, right panel). Therefore, Goal Maintenance neurons can achieve their role by altering the balance in the push-pull mechanism mediated by the direct (D1) and indirect (D2) pathways. This function is consistent with the observation that many of these neurons also exhibited higher activity after the feedback tone on unrewarded trials, previously described, and this activity influenced post-error adjustments in RT[17]. This is while the facilitation of response in Goal Maintenance neurons did not influence post-canceling changes in RT. Therefore, it is possible that these signals have different influences on their efferent targets depending on the task epoch.

We note that neurons with facilitated activity after SSRT were described in an investigation of the caudate nucleus of monkeys performing saccade countermanding[42]. The facilitation in the caudate nucleus coincided with that measured in SEF (Supplementary Fig. 9). The parallel between SEF and the striatum in patterns of modulation associated with proactive but not reactive inhibition is surprisingly, but satisfyingly, clear.

## Event-related potentials

We showed that macaque monkeys exhibit an N2/P3 ERP complex homologous to humans[7]. Disagreement persists about the frontal N2 and P3 index[76,77]. We found that the amplitude of the N2 during the stop-signal task varied most with the likelihood of error associated with experienced SSDs, and not conflict or SSD as previously suggested[7,78]. Consistent with previous reports of P3 indexing expectation and timing of behavior[76], we found that P3 amplitude co-varied most with the expected time of the feedback tone.

Variation in N2 and P3 polarization was predicted by the spiking of specific neuron classes in L2/3 and not L5/6. N2 magnitude was unrelated to the spiking of Goal Maintenance neurons but co-varied with the spiking of Conflict and Event Timing neurons, as well as the spiking of other neurons that did not modulate around the time of successful stopping. In contrast, P3 amplitude was predicted by the spiking of Goal Maintenance but not Conflict or Event Timing neurons. Also, N2 timing coincided with maximal modulation of Conflict and Event Timing neurons while P3, with Goal Maintenance neurons. A relationship between L2/3 spiking and these ERPs may appear trivial because the upper layers are closer to the surface EEG electrodes, but the result merits attention for several reasons. First, action potentials are too brief to generate the EEG, so the association with L2/3 spiking entails an association with coherent synaptic potentials[79]. Second, EEG polarization is related to the strength and orientation as much as the proximity of a dipole, and biophysical models of EEG sources assume that the larger L5 pyramidal neurons are the major contributor[46]. Finally, because ERPs arise from multiple sources, a dipole in one region can be canceled by a dipole oppositely oriented in another region. Therefore, it is unlikely that these L2/3 neurons are directly

causing these ERPs. It is more likely that the same synaptic potentials that result in the activation of these L2/3 neurons are also generating the EEG. Our results establish the N2 and P3 as possible biomarkers of the activity of neurons in the upper layers of SEF serving executive control functions. These results indicate that cognitive ERPs reflect diverse neuro-computational processes, rendering unitary and exclusive hypotheses incomplete.

## Incidence and multiplexing of signals

Here and in our previous report, we distinguished specific categories of neural signals. Different numbers of units were sampled in each category. Knowing that neural sampling with extracellular recording is biased in various ways, we cannot infer with confidence the importance of a process or the magnitude of computational contribution based on the number of units sampled. However, we will consider the reliability of the categories and the mixture of signals produced by single neurons.

The three signals reported here were multiplexed in single neurons with previously reported error and reinforcement signals[17]. That is, some neurons produced one kind of functional signal around the time of successful stopping and another at the time of reinforcement. Such multiplexing has been observed previously and can appear for several reasons[80,81]. First, modulation patterns may be too weak and variable to distinguish classes of neurons. Our selection criteria for neurons to analyze avoided this by including only neurons with distinct patterns of modulation. Second, frontal lobe neurons support diverse inputs and outputs from multiple cortical areas and subcortical nuclei. Therefore, a neuron can participate in partially overlapping but distinct networks such that in one state neurons broadcast one signal to some efferent targets, while in another state they broadcast another signal to other efferent targets. Theories about mixed-selectivity and dynamical systems have emphasized state-dependent dynamics[6,80], but they have not incorporated the specificity of laminar properties derived from specialized connectivity. Third, our classification of signals was based on response dynamics around the time of successful stopping, but we know of no theoretical or empirical prohibition against neurons modulating in association with multiple events. Ultimately, different neurons in different layers receive different inputs and have different outputs. Therefore, understanding the laminar distribution of signals reported in this study is a necessary step toward formulating more specific hypotheses about how neural networks function[82].

In conclusion, the present results add to the first catalog of the laminar functional architecture of an agranular frontal lobe area. Pioneering insights into the microcircuitry and mechanisms of the primary visual cortex began by describing the properties of neurons in different layers[18]. Contrasts of the results reported here with primary sensory areas will reveal the degree of computational uniformity across cortical areas. Being a source contributing to ERPs indexing performance monitoring and executive control, details about laminar processing in SEF offer unprecedented insights into the microcircuitry of executive control. These results validate the tractability of formulating neural mechanism models of performance monitoring and executive control, especially when constrained by formal[26], algorithmic[29,30], and spiking network[51] models of performance of a task with clear clinical relevance[83].

## Methods
### Animal care and surgical procedures

Data was collected from one male bonnet macaque (Eu, *Macaca radiata*, 8.8 kg, 6 y.o.) and one female rhesus macaque (X, *Macaca mulatta*, 6.0 kg, 8 y.o.) performing a countermanding task[21,25]. All procedures were in accordance with the National Institutes of Health Guidelines, the American Association for Laboratory Animal Care Guide for the Care and Use of Laboratory Animals and approved by the Vanderbilt Institutional Animal Care and Use Committee in accordance with the United States Department of Agriculture and Public Health Service policies. Surgical details have been described previously[84]. Briefly, magnetic resonance images (MRIs) were acquired with a Philips Intera Achieva 3 T scanner using SENSE Flex-S surface coils placed above or below the animal's head. T1-weighted gradient-echo structural images were obtained with a 3D turbo field echo anatomical sequence (TR = 8.729 ms; 130 slices, 0.70 mm thickness). These images were used to ensure Cilux recording chambers were placed in the correct area. Chambers were implanted normal to the cortex (Monkey Eu: 17°; Monkey X: 9°; relative to stereotaxic vertical) 1 mm right of the midline, 30 mm (Monkey Eu) and 28 mm (Monkey X) anterior to the interaural line.

### Acquiring EEG

EEG was recorded from the cranial surface with electrodes located over the MFC. Electrodes were referenced to linked ears using ear-clip electrodes (Electro-Cap International). The EEG from each electrode was amplified with a high-input impedance head stage (Plexon) and bandpass filtered between 0.7 and 170 Hz. Trials with blinks within 200 ms before or after the analysis interval were removed.

### Cortical mapping and electrode placement

Chambers implanted over the medial frontal cortex were mapped using tungsten microelectrodes (2–4 MΩ, FHC, Bowdoin, ME) to apply 200 ms trains of biphasic micro-stimulation (333 Hz, 200 μs pulse width). The SEF was identified as the area from which saccades could be elicited using <50 μA of current[85,86]. In both monkeys, the SEF chamber was placed over the left hemisphere. The dorsomedial location of the SEF makes it readily accessible for linear electrode array recordings across all cortical layers. A total of five penetrations were made into the cortex—two in monkey Eu, and three in monkey X. Three of these penetration locations were perpendicular to the cortex. In monkey Eu, the perpendicular penetrations sampled activity at site P1, located 4 mm lateral to the midline and 31 mm anterior to the interaural line. In monkey X, the perpendicular penetrations sampled activity at sites P2 and P3, located 4 mm lateral to the midline and 29 and 30 mm anterior to the interaural line, respectively.

### Acquiring neural spiking

Spiking activity and local field potentials were recorded using a 24-channel Plexon U-probe with 150 μm between contacts, allowing sampling from all layers. The U-probes were 100 mm in length with 30 mm reinforced tubing, 210 μm probe diameter, 30° tip angle, with 500 μm between the tip and first contact. Contacts were referenced to the probe shaft and grounded to the headpost. We used custom-built guide tubes consisting of 26-gauge polyether ether ketone (PEEK) tubing (Plastics One, Roanoke, VA) cut to length and glued into 19-gauge stainless steel hypodermic tubing (Small Parts Inc., Logansport, IN). This tubing had been cut to length, deburred, and polished so that they effectively support the U-probes as they penetrated the dura and entered the cortex. The stainless-steel guide tube provided mechanical support, while the PEEK tubing electrically insulated the shaft of the U-probe, and provided an inert, low-friction interface that aided in loading and penetration.

Microdrive adapters were fit to recording chambers with <400 μm of tolerance and locked in place at a single radial orientation (Crist Instruments, Hagerstown, MD). After setting up hydraulic microdrives (FHC, Bowdoin, ME) on these adapters, pivot points were locked in place by means of a custom mechanical clamp. Neither guide tubes nor U-probes were removed from the microdrives once recording commenced within a single monkey. These methods ensured that we were able to sample neural activity from precisely the same location relative to the chamber on repeated sessions.

Electrophysiology data were processed with unity-gain high-input impedance head stages (HST/32o25-36P-TR, Plexon). Spiking data were bandpass filtered between 100 Hz and 8 kHz and amplified 1000 times with a Plexon preamplifier, filtered in software with a 250 Hz high-pass filter, and amplified an additional 32,000 times. Waveforms were digitized at 40 kHz from −200 to 1200 μs relative to voltage threshold crossings. Thresholds were typically set at 3.5 standard deviations from the mean. All data were streamed to a single data acquisition system (MAP, Plexon, Dallas, TX). Time stamps of trial events were recorded at 500 Hz. Single units were sorted online using a software window discriminator and refined offline using principal components analysis implemented in Plexon offline sorter.

## Cortical depth and layer assignment

The retrospective depth of the electrode array relative to gray matter was assessed through the alignment of several physiological measures. Firstly, the pulse artifact was observed on a superficial channel which indicated where the electrode was in contact with either the dura mater or epidural saline in the recording chamber; these pulsated visibly in synchronization with the heartbeat. Secondly, a marked increase of power in the gamma frequency range (40–80 Hz) was observed at several electrode contacts, across all sessions. Previous literature has demonstrated elevated gamma power in superficial and middle layers relative to deeper layers[87]. Thirdly, an automated depth alignment procedure was employed which maximized the similarity of CSD profiles evoked by passive visual stimulation between sessions[21]. Further details can be found in ref. 21.

Further support for the laminar assignments was provided by an analysis of the depths of SEF layers measured in histological sections visualized with Nissl, neuronal nuclear antigen (NeuN), Gallyas myelin, acetylcholinesterase (AChE), non-phosphorylated neurofilament H (SMI-32), and the calcium-binding proteins parvalbumin (PV), calbindin (CB), and calretinin (CR)[17,21]. Additional information about the laminar structure was assessed through the pattern of cross-frequency phase-amplitude coupling across SEF layers[23]. Owing to variability in the depth estimates and the indistinct nature of the L6 border with white matter, some units appeared beyond the average gray-matter estimate; these were assigned to the nearest cellular layer.

## Acquiring eye position

Eye position data were collected at 1 kHz using an EyeLink 1000 infrared eye-tracking system (SR Research, Kanata, Ontario, Canada). This was streamed to a single data acquisition system (MAP, Plexon, Dallas, TX) and combined with other behavioral and neurophysiological data streams.

## Data collection protocol

Visual stimulus generation, event timing, and task control were done with TEMPO (version 16.12, set 49.20; Reflective Computing). The same protocol was used across monkeys and sessions. In each session, the monkey sat in an enclosed primate chair with its head restrained 45 cm from a CRT monitor (Dell P1130, background luminance of 0.10 cd/m²). The monitor had a refresh rate of 70 Hz, and the screen subtended 46° × 36° of the visual angle. After advancing the electrode array to the desired depth, they were left for 3–4 h until recordings stabilized across contacts. This led to consistently stable recordings with single units typically held indefinitely. Once these recordings stabilized, an hour of resting-state activity in near-total darkness was recorded. This was followed by the passive presentation of visual flashes followed by periods of total darkness in alternating blocks. Finally, the monkey performed approximately 2000 trials of the saccade countermanding (stop-signal) task per session.

## Countermanding task

The countermanding (stop-signal) task utilized in this study has been widely used previously[1]. Briefly, trials were initiated when monkeys fixated on a central point. Following a variable time period, drawn from an aging function to avoid anticipation of the visual stimulus[53], the center of the fixation point was removed leaving an outline. Simultaneously, a peripheral target was presented to the left or right of the screen.

On no-stop-signal trials, the monkey was required to shift its gaze to the target. Fixation on the target was required for 600 ms until an auditory tone sounded, whereupon monkeys could shift their gaze anywhere. The fluid reward was delivered 600 ms later. On stop-signal trials, comprising less than half of all trials, the center of the fixation point was re-illuminated after a variable stop-signal delay (SSD). An initial set of SSDs, typically separated by ~50 ms for Monkey Eu (45 ± 15 ms) and by ~100 ms for monkey X (115 ± 17 ms), were selected for each recording session. The selection of SSDs was adjusted to the idiosyncrasies of each subject to ensure performance satisfying key criteria for the stop-signal tasks. Different SSD values were used for the two subjects to account for between-subject differences in stopping performance[1]. To ensure that monkeys failed to countermand on ~50% of stop-signal trials, SSD was adjusted through an adaptive staircasing procedure. When a monkey failed to inhibit a response, the SSD was decreased by 1, 2, or 3 steps (randomly drawn) to increase the likelihood of success on the next stop trial. When a monkey canceled the saccade, SSD was increased by 1, 2, or 3 steps (randomly drawn) to decrease the likelihood of success on the next stop trial. On stop-signal trials, the monkey was required to maintain fixation on the central point until the tone sounded, whereupon monkeys could shift gaze anywhere. The fluid reward was delivered 600 ms later. By design, the duration from the target presentation until the tone was a fixed interval of 1500 ms. Thus, as SSD increased, the duration of fixation decreased (Fig. 1a).

Performance on this task is characterized by the probability of not canceling a saccade as a function of the SSD (the inhibition function) and the distribution of latencies of correct saccades in no-stop-signal trials and of noncanceled error saccades in stop-trials. Performance of the stop-signal task is explained as the outcome of a race between a GO and a STOP process[26]. The race model provides an estimate of the duration of the covert STOP process, and the time taken to accomplish response inhibition, known as stop-signal reaction time (SSRT)[29,30,51]. SSRT was calculated using two approaches—the conventional weighted-integration method and the more recent Bayesian Ex-Gaussian Estimation of Stop-Signal RT distributions (BEEST)[88] (Supplementary Figs. 4a, 5a, 6a). Compared to the weighted integration method, the Bayesian approach provides estimates of the variability in SSRT and the fraction of trigger failures for a given session[88]. Individual parameters were estimated for each session. Based on previous implementations of this approach, the priors were bounded uniform distributions ($\mu_{Go}$, $\mu_{Stop}$: $U$ (0.001, 1000); $\sigma_{Go}$, $\sigma_{Stop}$: $U$ (1, 500) $\tau_{Go}$, $\tau_{Stop}$: $U$ (1, 500); pTF: $U$ (0,1)). The posterior distributions were estimated using Metropolis-within-Gibbs sampling ran multiple through three chains. We ran the model for 5000 samples with a thinning of 5. None of our conclusions depend on the choice of SSRT calculation method. Full summary statistics of this analysis can be found in the data repository.

## Analysis of EEG

Methods paralleling those used in human studies were used. The N2 and P3 were obtained from average EEG synchronized on stop-signal presentation. Peak N2 was the time when the mean ERP reached maximal negativity in a 150–250 ms window after the stop-signal. Peak P3 was the time when the mean ERP was in a 250–400 ms window after the stop-signal. The amplitude of the N2 and P3 was quantified as the mean Z-transformed voltage for each SSD in a ± 50 ms window around

the maximal ERP deflection determined for each session. Indistinguishable results were obtained with wider (±75 ms), and narrower (±25 ms) windows or just the instantaneous maximal polarization. To characterize the polarizations associated with response inhibition, a difference ERP (ΔERP) was obtained by subtracting from the ERP recorded on canceled trials the ERP recorded on RT-matched no-stop-signal trials.

## Analysis of neural spiking

Spike widths exhibited a bimodal distribution (Supplementary Fig. 1a), and neurons were distinguished as narrow- (peak-to-trough duration ≤ 250 μs) or broad-spikes (>250 μs). Measurements of neural spiking were based on spike density functions (SDF) produced by convolving the spike train with a kernel resembling a postsynaptic potential defined by

$$\text{SDF}(t) = \left(1 - \exp\left(\frac{-t}{\tau_\text{g}}\right)\right) \times \exp\left(\frac{-t}{\tau_\text{d}}\right) \qquad (3)$$

with growth time constant ($\tau_\text{g}$) of 1 ms, and decay time constant ($\tau_\text{d}$) of 20 ms, corresponding to the values measured for excitatory postsynaptic potentials. The area of the kernel was set to equal 1. To analyze spiking activity associated with successful stopping, we compared the activity on canceled trials and on no-stop-signal trials with RT greater than SSD + SSRT. This latency-matching compares trials in which countermanding was successful with trials in which countermanding would have been successful had the stop signal been presented. Neurons were distinguished by patterns of modulation consisting of periods of facilitation or suppression using a consensus clustering algorithm[28] (Fig. 1c, Supplementary Fig. 1c–e). The input to this analysis pipeline was the SDF on canceled trials and on latency-matched no-stop-signal trials during the 100 ms preceding SSRT and 200 ms following SSRT. Results did not change much if interval durations were changed.

To prevent outlying values from exerting excessive influence, population spike density plots were obtained by scaling the SDF of each neuron by the 95% confidence interval between the 2.5% lowest rate and the 97.5% highest rate in one of two intervals. The first interval was a 600 ms window centered on SSRT on canceled and on no-stop-signal trials. The second interval was −100 to +300 ms relative to the feedback tone.

To identify spiking modulation, we applied methods previously employed. First, we calculated a difference function (ΔSDF), the difference between the SDF on canceled and latency-matched no-stop-signal trials. Periods of statistically significant modulation were identified based on multiple criteria—(a) the difference function must exceed by at least 2 standard deviations a baseline difference measured in the 100 ms interval before the target appeared, (b) the difference must occur from 50 ms before to 900 ms after the stop-signal, and (c) the difference must persist for at least 100 ms (or for 50 ms if the difference exceeded baseline by 3 standard deviations). As commonly found in the MFC, some neurons exhibited low spiking rates. To obtain reliable estimates of modulation times, we also convolved the SDF with a square 8 ms window. The modulation intervals were validated by manual inspection.

To determine modulation associated with the systematically variable timing of the feedback tone on canceled trials, the SDF was compared against the minimum value found between 500 ms before and 900 ms after the tone. Focusing on modulation occurring only during the period of operant control on behavior, modulations beginning less than 300 ms after the tone were not included. For comparisons across neurons and sessions, Z-transformed SDF or ΔSDF was used.

## Mixed-effects models

We fit variation in spike counts or EEG voltage to models of task events and performance outcomes (Fig. 1, Supplementary Table 2, Supplementary Fig. 3). To determine which performance measure accounted best for the variation of neural measures, the performance and neural quantities were averaged within groups of early-, mid-, and late-SSD trials. SSD values greater than ~350 ms were not included because too few canceled trials were obtained. The analysis of the facilitation after SSRT was based on ΔSDF (Figs. 3 and 5), but the major conclusions held if the analysis used SDF. The analysis of the modulation before SSRT or the feedback tone (Fig. 4) was based on the SDF of canceled trials. Before SSRT the SDF of canceled and no-stop-signal trials was not different. Before the feedback tone, the interval was variable on canceled trials but not on no-stop-signal trials, and longer on canceled relative to no-stop-signal trials.

Each model was defined by one parameter. A limited number of SSDs (typically between 3–7, divided into 3 bins) offered sufficient trials for analysis. Because of the dependence of the tested parameters on SSD, the values for each parameter varied with SSD. Although performance and behavioral parameters could be correlated, the non-linear relations between them and session-wise variations empowered effective model comparison (Supplementary Fig. 3b).

Mixed-effects models of ΔSDF, SDF, or ΔERP values in relation to the various performance measures were compared using Bayesian Information Criteria (BIC), which provides a criterion for model selection. We report the results of the most basic version of each model with the main effect term corresponding to the performance parameter and random intercepts grouped by a neuron (for spiking) or session (for EEG). The values for each performance parameter were z-transform normalized for a fair comparison between models related to different quantities. All models had the same degrees of freedom, allowing direct comparison of BIC values between models. The smallest BIC identified the best model. ΔBIC (BIC$_\text{best}$−BIC$_\text{competing}$) quantified the fit of the other models relative to the best with ΔBIC < 2 were considered as candidate models, 2 < ΔBIC < 6 earning weak support, and ΔBIC > 6 supporting rejection of the model[89,90]. Models with non-significant main effects were also rejected. More complex versions of these models resulted in similar conclusions. Mixed-effects models were performed using MATLAB's Statistical Toolbox.

## Relating RT and neural spiking

All neurons were tested. For facilitated neurons, we counted spikes immediately following SSRT, within the neuron-specific modulation window. For suppressed neurons, we counted spikes in a 20 ms interval centered on the peak of the ramp prior to the suppression. Because both spiking activity and RT varied with SSD, a multiple linear regression model controlled for SSD to test whether spike rates varied with RT.

## Relating N2/P3 and neural spiking

We used the method described previously to establish the relationship between spiking activity and the ERN[17]. Single-trial spiking was the mean convolved spike data for that trial recorded from neurons in L2/3 and in L5/6 of perpendicular penetrations within ±50 ms of the N2 and P3 peaks. To account for variations in ERP voltage and spike counts across sessions, a fixed-effects adjustment was performed by centering each distribution on its mean and dividing by its most extreme value. To measure the N2/P3 amplitudes robustly, we grouped rank-ordered single-trial ERP values into 20 successive bins. From trials in each bin, we calculated the mean N2 and mean P3 magnitude (dependent variable), the mean spike count in the upper and lower layers (independent variables), and the average SSD, on Canceled trials. Data from all sessions were combined for a pooled partial correlation. Each point in Fig. 6 plots the paired values of the mean normalized ERP voltage and normalized activity for each of the 20 bins from every session. The

statistical relationship between ERP magnitude and spiking activity was quantified through multiple linear regression on normalized data pooled across sessions. Two factors were considered: (1) spiking activity in L2/3, and (2) spiking activity in L5/6. Both spiking activity and ERPs also varied with SSD but the inclusion of SSD as a predictor did not change the results.

## Reporting summary

Further information on research design is available in the Nature Research Reporting Summary linked to this article.

## Data availability

Other aspects of the behavioral and neural dataset have been published previously[17,21,23]. The processed data used in this study are openly available online through the Open Science Framework (OSF). https://doi.org/10.17605/OSF.IO/3RZJX. Available online via: https://osf.io/3rzjx/. The raw data analyzed in the current study are available from the corresponding author on reasonable request.

## Code availability

The analysis codes used in this study are openly available online through the Open Science Framework (OSF). https://doi.org/10.17605/OSF.IO/3RZJX. Available online via: https://osf.io/3rzjx/.

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

## Acknowledgements
The authors thank D. Godlove for data collection, G. Luppino, M. Matsumoto, N. Palomero-Gallagher, and L. Rapan for sharing data; J. Elsey, M. Feurtado, M. Maddox, S. Motorny, J. Parker, D. Richardson, M. Schall, C.R. Subraveti, L. Toy, B. Williams, and R. Williams for animal care and other technical assistance; and Z. Fu, M. Matsumoto, P. Redgrave, U. Rutishauser, E. Sigworth, A. Tomarken, and G. Woodman for helpful discussions. Imaging data were collected at the Vanderbilt Institute of Imaging Science. This work was supported by R01-MH55806 (J.D.S.), R01-EY019882 (G. Woodman), P30-EY08126 (J.D.S. and D. Calkins), NSERC RGPIN-2022-04592 (J.D.S.), CIHR Post-Doctoral Fellowship (A.S.), by Robin and Richard Patton through the E. Bronson Ingram Chair in Neuroscience, and by the Vision Science to Application (VISTA) program at York University.

## Author contributions
Experimental design, J.D.S. Data collection, J.D.S. Data analysis, A.S. and S.E. Interpretation and preparation of the manuscript, A.S., S.E., and J.D.S.

## Competing interests
The authors declare no competing interests.
