## [Peer Review File · Nature Communications]

Functional architecture of executive control and associated event-related potentials in macaquesREVIEWER COMMENTS

Reviewer #1 (Remarks to the Author):

Review – Sajad et al. Functional architecture of executive control and associated event-related potentials.

The manuscript by Sajad and colleagues describes a study in which the authors sought to extend existing knowledge of the contribution of supplementary eye fields (SEF) to performance monitoring and adaptive control by carrying out laminar recordings during a modified version of the well-studied saccade countermanding paradigm to characterize the response properties of single SEF neurons across cortical layers. Event-related potentials were also recorded at the cortical surface and the contribution of single neurons to the ERP N2/P3. They describe several patterns of response properties during the task, and their laminar distributions: 1) Conflict neurons, modulated after the SSRT in a manner consistent with coactivation of GO and STOP processes, distributed across all cortical layers, 2) Time Keeping neurons, modulated in a manner consistent with representation of timing of task events and interval durations. A subset of these neurons exhibiting ramping activity showed some laminar differentiation, although in general this type was found across all layers, and 3) Goal Maintenance neurons, revealed by the modification of the task used here which required the animals to maintain fixation on the visual stop signal until a tone signaled that a saccade to the target was permitted to obtain reward. These neurons exhibited sustained activity during this time period consistent with short-term retention of the task goal. In contrast to the other two neuronal types, Goal Maintenance neurons were found most commonly in layers 2/3 rather than layers 5/6, and more of these neurons were characterized as narrow spiking putative interneurons. Finally, the authors also observed an N2/P3 ERP component on successful stopping trials, consistent with findings in human participants. N2 polarization was found to be predicted by spiking activity in layers 2/3 but not 5/6. P3 polarization was predicted by the spiking activity of goal maintenance neurons in layers 2/3 but not 5/6.

Overall, this is an excellent manuscript and the data represent a significant and timely advance. The laminar distributions and broad and narrow-spiking classifications together with knowledge of the response profiles during the countermanding task are new contributions, and are critical for advancing knowledge and developing models of cortical microcircuits and their contributions to cognitive processes more generally. Indeed, a very detailed model of this sort is presented in the discussion of the manuscript. The authors employ a well-established and thoroughly modeled behavioral paradigm requiring response inhibition and response monitoring, which has been studied extensively in human participants as well as non-human primates and thus has much value in the translation of findings. The addition of the ERP component to the study makes this particularly so, since an electrophysiological link between the two species is established as well. Studies of this sort are of immeasurable value in understanding the neural basis of response inhibition and cognitive control.

The quality of the data are excellent, and all analyses are thorough and appropriate. The laboratory of the senior author is expert in the countermanding paradigm, the SEF, and all other procedures employed here. I am happy to recommend publication of the manuscript with no major revisions.

I have only one minor comment with respect to the Goal Maintenance neurons described here. The authors ascribe the sustained activity they observed to a mnemonic process representing the task goal. I wonder if there were any trials on which the animals broke fixation prior to the tone and whether this type of activity was absent in this case ?

Reviewer #2 (Remarks to the Author):

Summary

The authors recorded EEG and neural spiking over and in the supplementary eye field (SEF) of macaques during a saccade countermanding task, to better understand the neural circuitry and laminar structure that implements executive control and eventually broadcasts event-related potentials (ERPs) to the brain surface. They found functionally different neuronal groups important for various task elements such as conflict monitoring, time keeping and goal maintenance. Particular N2 and P3 ERP components were predicted by the different neuronal groups in different cortical layers. These findings provide valuable information about the origins of scalp-recorded ERPs, and demonstrate how to combine multiple scales of electrophysiology (EEG, single-unit spiking activity).

Major concerns/criticisms

- Line 121-123, 167, 398-401, 912-914, 921-923. It's not clear why activity on canceled trials is compared to no stop-signal trials if you want to find out what is associated with successful response inhibition. The comparison between canceled and non-canceled trials is ignored in these cases, but is relevant for studying successful response inhibition, as this comparison shows the inhibition-specific activity. As the authors decided to leave this comparison out in almost the entire manuscript it needs proper justification why this comparison is not relevant according to them. Nevertheless, my suggestion would be to at least include this comparison in the main analyses and add the non-canceled trials to the figures as suggested in the minor concerns/criticisms.

- I'm confused about how $p(\text{error})$ is computed. Is this calculated per-session? Or over a running history of the previous N trials? Or per RT bin?

- I was surprised that the authors did not compute the N2/P3 in the silicon probe. Isn't that the best intermediate step in between the spikes and the EEG? According to the methods section, the LFP data were recorded.

- The authors equate $p(\text{error})$ with conflict, but 'conflict' in the cognitive control literature typically indicates a competition between two (or more) different responses, with one being automatic and the other being task-driven (as typified by Stroop, Simon, and flanker tasks). The stop-signal task has only one action at a time, which is fully engaged or canceled. Clearly this task requires cognitive control and inhibition, but it's not the same kind of conflict as in many other tasks. (It's also worth noting here that the Yeung/Botvinick model of conflict= $a_1 * a_2$ has not held up to empirical scrutiny.) I suggest to rename this to something else, e.g., "p(error) neurons" or "difficulty-encoding neurons." I realize that the authors might disagree (cf Schall and Boucher 2007), but at the least, a serious discussion needs to be added that the term 'conflict' is used here in a different way than other parts of the literature.

Relatedly, I don't mean to be too pedantic here, but increased errors are not necessarily coupled to increased conflict. Errors rates also vary by time pressure, task complexity, working memory load, attentional demands, time-on-task and other factors. The point is that calling neurons whose activity varies with $p(\text{error})$ 'conflict neurons' seems inappropriately speculative for the Results section. I guess they used this cognitive label to link to the N2, but that seems more a Discussion section point than a Results section label.

Minor concerns/criticisms

- In Figure 2a, 3a, 4a and 5a I would like to also see the line for the non-canceled trials. It's easy put them in and gives a more complete view of the contrasts between the different trial types. The contrast between canceled and non-canceled trials is relevant, as it shows which part of activity is specific to the successful execution of countermanding. This will also make the figure more convincing, as now it seems like the authors hide these non-canceled trials for an unwanted reason.

- In the first paragraph of the Discussion section the authors refer to Figure 6. However, I prefer the explanation of this figure to be embedded in the Discussion instead of having an entirely separate paragraph in the figure description. This will make it more of a complete single story, instead of having

a separate story in the figure description.

- Line 876-878. It is unclear why the set of SSDs for the different monkeys have different separation sizes (40-60 ms vs. 100 ms). This needs explanation.
- Wouldn't it be easier to call 'no stop signal' trials 'go' trials? The authors use the term 'go' several other times in the manuscript.
- Figure 2c doesn't really seem to be a 'representative' neuron considering its peak responses is later than most of the neurons shown in Figure 2b. Is it possible to show more data? Perhaps examples of putative pyramidal and inhibitory cells that peak around SSRT vs later? Maybe you can highlight the specific 'representative' neuron by highlighting it in Figure 2b.
- Same comment for Figure 3c. In fact, from looking at Figure 3b, I cannot tell which neuron 3c comes from. Maybe you can highlight the specific neuron in Figure 3b, like suggested for Figure 2b.

Line edits

- Please have a native English speaker smooth out the manuscript. Some sentences read like notes rather than full sentences (e.g. line 53), and there are many minor mistakes that could be corrected, like missing articles (e.g. line 17, 40, 51, 606, 610).

Reviewer #3 (Remarks to the Author):

This paper is an important step toward a better understanding of cortical microcircuitry involved in executive control. As a continued effort to dissect laminar organization of medial frontal cortex during stop-signal task (Sajdad et al., Nat. Neuro. 2019), this paper focused on more critical trial types (canceled) and task epochs (around SSRT) and found interesting functional dissociations across different neuronal groups. In particular, it is noteworthy finding that "goal maintenance" narrow-spiking neurons are specifically localized in L2/3. Furthermore, the report of N2/N3 in the monkey model is novel and its relation to neural spiking activates is an important addition to the field. The paper has solid and convincing results. That said, I have some suggestions that can potentially improve the paper.

- Major points

More justification is needed for the approach of classifying individual neurons into distinct groups – conflict monitoring, event timing, goal maintenance. While it is justified in terms of previous findings (e.g., for conflict neurons) and behavioral demands, this classification seems arbitrary, post-hoc, and not based on first principles (i.e., derived from theory or model). Why are there three, not two or not four, categories? Even clustering results in Supplementary figure 1 do not speak to this issue.

Another way to ask this question is, how should readers interpret the different number of neurons in each group? Does it matter? Because there is a relatively smaller number of goal-maintenance neurons, should we conclude that SEF is involved more in conflicting monitoring and time keeping? Or does SEF do everything including the processing of gain and loss (Sajdad et al., Nat. Neuro. 2019)? How these different groups of neurons interact and what is the population-level understanding? I know this is a high-level question that may need brand-new analyses (like population-level decoding or state-space/dynamical-systems approach). But at least, this issue should be thoroughly discussed as medial frontal cortex has notoriously heterogeneous neural populations in their sensorimotor selectivity.

Another key question is, is this classification a distinct category or more like a continuum? The latter is more likely given their overlapped dynamics (Fig. 5b, Sup. Fig. 1e). The difference between conflict and maintenance neurons is only about how long the neurons sustain their activities (e.g., transient versus persistent) and so not clear cut at all. Specifically, maintain neurons can be well considered as conflict monitoring and vice versa. This issue seems to become more problematic as even predictions between models are similar – as mentioned in methods, model predictions and parameters are correlated (e.g., SSD, $p(\text{error})$, conflict, and time to tone) and it is not clear how "random variations"

(across single trials?) can differentiate these models.

Along the line of the previous comment, model comparison results should be presented better. In particular, details of alternative models are buried in the text/supplementary and hard to digest. It'd be useful to have a figure showing predictions from different models (well explained in method, "mixed-effects model") and to bring a simplified version of Sup. Table 2 into the main manuscript.

Another big question is, what is the behavioral relevance of these activities in stop-signal task. More specifically, the behavioral relevance of conflict monitoring is not directly tested. In discussion, it is mentioned that conflict monitoring can be useful for post-stopping slowing but this interesting idea is not tested at all. For instance, was the go RT slower in the next trial after the conflict-monitoring neurons showed a higher firing rate? Does animal use timing information in 'time-keeping neurons' to predict SSD and adjust behavior accordingly?

Given that no specificity was found for conflict neurons in terms of laminar and spiking width (χ^2 analyses), the conclusion that those neurons are broad-spiking and in L3/5 (in abstract, Line 23-25) is not warranted. This is just a by-product of sampling neurons in SEF. The valid conclusion is that no specificity was found.

Motivation for classifying neurons into broad-spiking and narrow-spiking is also not clear. It'd be useful to provide backgrounds or hypotheses in the introduction. For example, is there any previous anatomical study suggesting inhibitory interneurons are more common in L2/3? It seems to be the case in Supp. Fig. 1.

It'd be useful to explicitly state the null hypothesis when linking N2/P3 and spiking activities across layers. Is it somewhat obvious that ERPs would be better predicted by upper layers as they are closer? Also, due to their match in timing, isn't it obvious that monitoring/timing neurons better predict N2 and maintenance neurons better predict P3? One way to tackle the former question is to test how well the cranial EEG is predicted by LFP signals across layers.

- Minor points

In figure 2,3, and 4, the bottom plots in panel (a) have a confusing label, P(active). Because p(active) is different between left and right panels, it'd be better to use a separate y-axis label.

I can easily imagine, to readers naïve to the stop-signal task, the paper is written a bit difficult to follow and jargon-heavy fashion. If it is written without assuming prior knowledge, the paper will receive a wider readership.

Y-axis Label of the right panel in fig. 3c should be pre-SSRT, to be consistent with left panel shades and legends.

Why did conflict neurons show higher activity right before feedback tone (Fig. 2a, Sup. Fig. 3b)?

REVIEWER COMMENTS

Reviewer #1 (Remarks to the Author):

Review – Sajad et al. *Functional architecture of executive control and associated event-related potentials.*

The manuscript by Sajad and colleagues describes a study in which the authors sought to extend existing knowledge of the contribution of supplementary eye fields (SEF) to performance monitoring and adaptive control by carrying out laminar recordings during a modified version of the well-studied saccade countermanding paradigm to characterize the response properties of single SEF neurons across cortical layers. Event-related potentials were also recorded at the cortical surface and the contribution of single neurons to the ERP N2/P3. They describe several patterns of response properties during the task, and their laminar distributions: 1) Conflict neurons, modulated after the SSRT in a manner consistent with coactivation of GO and STOP processes, distributed across all cortical layers, 2) Time Keeping neurons, modulated in a manner consistent with representation of timing of task events and interval durations. A subset of these neurons exhibiting ramping activity showed some laminar differentiation, although in general this type was found across all layers, and 3) Goal Maintenance neurons, revealed by the modification of the task used here which required the animals to maintain fixation on the visual stop signal until a tone signaled that a saccade to the target was permitted to obtain reward. These neurons exhibited sustained activity during this time period consistent with short-term retention of the task goal. In contrast to the other two neuronal types, Goal Maintenance neurons were found most commonly in layers 2/3 rather than layers 5/6, and more of these neurons were characterized as narrow spiking putative interneurons. Finally, the authors also observed an N2/P3 ERP component on successful stopping trials, consistent with findings in human participants. N2 polarization was found to be predicted by spiking activity in layers 2/3 but not 5/6. P3 polarization was predicted by the spiking activity of goal maintenance neurons in layers 2/3 but not 5/6.

Overall, this is an excellent manuscript and the data represent a significant and timely advance. The laminar distributions and broad and narrow-spiking classifications together with knowledge of the response profiles during the countermanding task are new contributions, and are critical for advancing knowledge and developing models of cortical microcircuits and their contributions to cognitive processes more generally. Indeed, a very detailed model of this sort is presented in the discussion of the manuscript. The authors employ a well-established and thoroughly modeled behavioral paradigm requiring response inhibition and response monitoring, which has been studied extensively in human participants as well as non-human primates and thus has much value in the translation of findings. The addition of the ERP component to the study makes this particularly so, since an electrophysiological link between the two species is established as well. Studies of this sort are of immeasurable value in understanding the neural basis of response inhibition and cognitive control.

The quality of the data are excellent, and all analyses are thorough and appropriate. The laboratory of the senior author is expert in the countermanding paradigm, the SEF, and all other procedures employed here. I am happy to recommend publication of the manuscript with no major revisions.

We thank the reviewer for their kind comments and appreciation of the manuscript.

I have only one minor comment with respect to the Goal Maintenance neurons described here. The authors ascribe the sustained activity they observed to a mnemonic process representing the task goal. I wonder if there were any trials on which the animals broke fixation prior to the tone and whether this type of activity was absent in this case?

Thank you for this idea. Indeed, we had examined this but had not included it in the manuscript because the number of useful aborted canceled trials was too small to support a confident conclusion. However, because both reviewers 1 and 3 have asked this question, we are now showing these results in **Supplementary Figure 6f** and report the finding in the Results section, **Page 11, Lines 328-330**.

We examined the activity of neurons on trials in which fixation was successfully maintained (i.e., canceled trials) against trials in which the fixation was broken by a saccade or a blink (i.e., aborted) in a period between SSRT and feedback tone. Because we had not conceived of analyzing aborted trials, on these trials only the TrialStart event was saved. So, we reconstructed the timing of saccades, fixation breaks, and blinks that aborted the trials relative to this time. We included only a subset of the aborted trials in which we identified a saccade or blink confidently during the period between SSRT and feedback tone, and the proportions of trials were matched

for SSD. Only 14/54 Goal Maintenance neurons had ≥ 5 trials for meaningful comparison of neural activity. These neurons showed reduced activity on fixation break trials compared to successful stop trials.

We recently completed sampling neural activity from two other monkeys doing this task. As shown in the figure below, we replicated the basic observations of these neurons.

Reviewer #2 (Remarks to the Author):*Summary*

The authors recorded EEG and neural spiking over and in the supplementary eye field (SEF) of macaques during a saccade countermanding task, to better understand the neural circuitry and laminar structure that implements executive control and eventually broadcasts event-related potentials (ERPs) to the brain surface. They found functionally different neuronal groups important for various task elements such as conflict monitoring, time keeping and goal maintenance. Particular N2 and P3 ERP components were predicted by the different neuronal groups in different cortical layers. These findings provide valuable information about the origins of scalp-recorded ERPs, and demonstrate how to combine multiple scales of electrophysiology (EEG, single-unit spiking activity).

We are pleased that the reviewer appreciated our contribution.

Major concerns/criticisms

- Line 121-123, 167, 398-401, 912-914, 921-923. It's not clear why activity on canceled trials is compared to no stop-signal trials if you want to find out what is associated with successful response inhibition. The comparison between canceled and non-canceled trials is ignored in these cases, but is relevant for studying successful response inhibition, as this comparison shows the inhibition-specific activity. As the authors decided to leave this comparison out in almost the entire manuscript it needs proper justification why this comparison is not relevant according to them. Nevertheless, my suggestion would be to at least include this comparison in the main analyses and add the non-canceled trials to the figures as suggested in the minor concerns/criticisms.

We appreciate this comment and have revised the text to clarify why the Logan race model of countermanding performance guides the trial comparisons that we used. In short, including the error non-canceled trials would be misleading for two reasons:

- (1) Erroneous non-canceled trials happen because the GO process was too fast producing shorter RT. Correct canceled trials happen because the GO process happened to be slow enough to be interrupted by the STOP process. This fundamental property is the basis of the Logan race model and is reflected clearly in the patterns of neural activity observed in motor structures (Hanes et al. 1998 J Neurophysiol; Paré & Hanes, 2003 J Neurosci). Consequently, comparing neural activity between non-canceled and canceled trials is confounded by differences in the visual and motor processes producing the earlier and the later parts of RT distributions. The standard approach in this literature is to “latency-match” non-canceled trials with no-stop trials with $RT < SSD + SSRT$ and canceled trials with no-stop trials with $RT > SSD + SSRT$. This approach is used in studies of spiking as well as ERPs (e.g., see Figure 2 in Kok, A., Ramautar, J. R., De Ruyter, M. B., Band, G. P., & Ridderinkhof, K. R. (2004). ERP components associated with successful and unsuccessful stopping in a stop-signal task. *Psychophysiology*, 41(1), 9-20).
- (2) As reported previously (Stuphorn et al. 2000; Sajad et al. 2019), neural activity in SEF on non-canceled trials has an additional error component that can confuse the interpretation (see **Supplementary Figure 2b**).

We have clarified the rationale for not including non-canceled trials in the Results (**Pages 4-5, Lines 95-105**) and in a new **Supplementary Figure 2**. Nevertheless, we now show the spiking activity on non-canceled trials in **Supplementary Figure 2b** for the 3 neuron classes described in this manuscript.

- I'm confused about how $p(\text{error})$ is computed. Is this calculated per-session? Or over a running history of the previous N trials? Or per RT bin?

We thank the reviewer for highlighting this ambiguity. We have thoroughly revised the text in the Results (**page 5, Lines 106-128**) and Methods (**page 28, Lines 886-890**) and explain the models in revised **Figure 1**. We also changed the nomenclature for the conflict model to $p(\text{NC} | \text{SSD})$ and

the error-likelihood model to $p(NC_{error} | SSD) / p(SS_{seen} | SSD)$. We did not calculate a running history of N trials, but we will look at this in the future. For the conflict model we combine data within each session based on the parameters of the inhibition function for each stop-signal delay bin. For the error likelihood model we combine data within each session for each stop-signal delay bin.

- I was surprised that the authors did not compute the N2/P3 in the silicon probe. Isn't that the best intermediate step in between the spikes and the EEG? According to the methods section, the LFP data were recorded.

Thank you for this suggestion. Certainly, the laminar LFPs offer an important perspective. In fact, this is the focus of an ongoing collaboration to describe the biophysical origin of the EEG signal. (Herrera B, Sajad A, Woodman GF, Schall JD, Riera JJ. A minimal biophysical model of neocortical pyramidal cells: Implications for frontal cortex microcircuitry and field potential generation. *J Neurosci*. 2020 Oct 28;40(44):8513-8529). Manuscripts are in preparation now characterizing the laminar patterns of LFP in SEF during this task. In parallel with Sajad et al. 2019, we restricted the scope of this manuscript to the relationship between EEG signals and concomitant laminar patterns of neural spiking. We have revised the text to clarify this distinction (**Page 13, Lines 394 - 396**).

- The authors equate $p(\text{error})$ with conflict, but 'conflict' in the cognitive control literature typically indicates a competition between two (or more) different responses, with one being automatic and the other being task-driven (as typified by Stroop, Simon, and flanker tasks).

*The stop-signal task has only one action at a time, which is fully engaged or canceled. Clearly this task requires cognitive control and inhibition, but it's not the same kind of conflict as in many other tasks. (It's also worth noting here that the Yeung/Botvinick model of $\text{conflict} = a_1 * a_2$ has not held up to empirical scrutiny.) I suggest to rename this to something else, e.g., "p(error) neurons" or "difficulty-encoding neurons." I realize that the authors might disagree (cf Schall and Boucher 2007), but at the least, a serious discussion needs to be added that the term 'conflict' is used here in a different way than other parts of the literature.*

We appreciate the nuanced understanding revealed by this comment and acknowledge that the literature has evolved from the simple "conflict = $a_1 * a_2$ " framework. We have revised the text and supplementary information to clarify and justify the utility of this framework in the description of these data. Our implementation of the concept of conflict flows directly from the interactive race model (Boucher L, Palmeri TJ, Logan GD, Schall JD. Inhibitory control in mind and brain: an interactive race model of countermanning saccades. *Psychol Rev*. 2007 Apr;114(2):376-97; Logan GD, Yamaguchi M, Schall JD, Palmeri TJ. Inhibitory control in mind and brain 2.0: blocked-input models of saccadic countermanning. *Psychol Rev*. 2015 Apr;122(2):115-47), which by the way applies to perceptual decision-making tasks with competing response tendencies (Middlebrooks PG, Zandbelt BB, Logan GD, Palmeri TJ, Schall JD. Countermanning perceptual decision-making. *iScience*. 2020 Jan 24;23(1):100777). As demonstrated in revised Figure 1 and Supplementary Figure 3, the GO and STOP units are co-active before and after SSRT, so "conflict = $a_1 * a_2$ " yields interpretable values. In Results and Discussion, we test alternative interpretations of surprise and salience and as the reviewer suggested acknowledge that this may be related to encoding difficulty (**Page 15, Line 440-472**).

We must observe, though, that many other investigators have used the stop-signal task to investigate conflict. Originally, Brown & Braver (2005, Learned predictions of error likelihood in the anterior cingulate cortex, *Science*, 307(5712):1118-21) used a modified stop signal task (stop-change or double-step task) to evaluate conflict and error detection and develop their error-likelihood theory of ACC function. We note that Braver was a co-author of Botvinick et al. 2001. Subsequently, Stahl & Gibbons (2007, Dynamics of response-conflict monitoring and individual differences in response control and behavioral control: an electrophysiological investigation using a stop-signal task. *Clin Neurophysiol*. 118(3):581-96), motivated by the Yeung, Botvinick & Cohen 2004 description of conflict in the interpretation of the ERN, investigated the ERN in a stop-signal task and observed, "However, independently of response type Ne/ERN also increased with

increasing stop-signal delay. Since longer delay invokes stronger response conflict, results specifically support the notion of Ne/ERN reflecting response-conflict monitoring". Similarly, Aron, Behrens, Smith, Frank, and Poldrack (2007, Triangulating a cognitive control network using diffusion-weighted magnetic resonance imaging (MRI) and functional MRI, *J Neurosci*, 27(14):3743-52) used a conditional stop-signal paradigm to examine behavioral and neural signatures of conflict-induced slowing. The comprehensive model of Wiecki & Frank (2013, A computational model of inhibitory control in frontal cortex and basal ganglia, *Psychol Rev*, 120(2):329-55) incorporates the conceptual framework that we are evaluating in this manuscript. Consider their statement of the problem, "In the current model, conflict is computed explicitly by the dorsal anterior cingulate cortex (dACC), which detects when multiple competing FEF response units are activated concurrently, and in turn activates the STN to make it more difficult to gate any response until this conflict is resolved. (page 332)". Finally, most recently, Kleinsorge (2021, Stimulus-response conflict tasks and their use in clinical psychology, *Int J Environ Res Public Health*, 18(20):10657) wrote, "Although the stop-signal task is not a typical stimulus-response conflict task, it induces conflict between a go- and a stop-signal that can be pitted against each other in a methodologically elegant way that has yielded important conceptual insight into the nature of 'automaticity'..." We trust that this brief review justifies conceptually our employment of this concept. We turn now to the empirical justification.

Relatedly, I don't mean to be too pedantic here, but increased errors are not necessarily coupled to increased conflict. Errors rates also vary by time pressure, task complexity, working memory load, attentional demands, time-on-task and other factors. The point is that calling neurons whose activity varies with p(error) 'conflict neurons' seems inappropriately speculative for the Results section. I guess they used this cognitive label to link to the N2, but that seems more a Discussion section point than a Results section label.

We understand the reviewer's point and agree that performance errors can arise for many different reasons. A utility of the stop-signal task is that most errors can be explained by the race of the GO and STOP processes. The original Botvinick measure of conflict is easily derived from the time course of activation of the GO and STOP process that are governing performance. That quantity happens to relate in an intelligible manner to the inhibition function, which characterizes error rate in this task. To clarify the presentation, we are happy to revise the nomenclature for the conflict model. We believe this change in nomenclature clarifies that we do not equate conflict simply with error and thus addresses the reviewer's very useful comment.

Minor concerns/criticisms

- In Figure 2a, 3a, 4a and 5a I would like to also see the line for the non-canceled trials. It's easy put them in and gives a more complete view of the contrasts between the different trial types. The contrast between canceled and non-canceled trials is relevant, as it shows which part of activity is specific to the successful execution of countermanding. This will also make the figure more convincing, as now it seems like the authors hide these non-canceled trials for an unwanted reason.

For reasons we detailed above, the direct comparison between canceled and non-canceled trials is not valid and can be misleading. However, in revised **Supplementary Figure 2b** we plot activity on non-canceled trials for comparison.

- Line 876-878. It is unclear why the set of SSDs for the different monkeys have different separation sizes (40-60 ms vs. 100 ms). This needs explanation.

We have explained this in the revised Methods section and cited relevant literature on **Page 25, Lines 797-798; Page 3, Line 57-58**. The key aspect of the stop-signal task is to have a set of SSDs that probe response inhibition low to high error rates. Monkeys, like people, are idiosyncratic, so different SSD values must be used.

- Wouldn't it be easier to call 'no stop signal' trials 'go' trials? The authors use the term 'go' several other times in the manuscript

We avoid the term to 'go' to avoid the following confusions:

- (1) In the literature 'go' trials refer to other manipulations and the saccade response in other conditions (e.g., GO/NO-GO task).
- (2) The term "no-stop signal trial" is consistent with the terminology used in the stop-signal task literature.
- (3) In this literature the term GO refers to the race model GO processes that results in the generation of the saccade. Conceptually, this is not equivalent to no-stop trials.

As requested by the reviewer we have now made the terminology consistent throughout the manuscript.

- Figure 2c doesn't really seem to be a 'representative' neuron considering its peak responses is later than most of the neurons shown in Figure 2b. Is it possible to show more data? Perhaps examples of putative pyramidal and inhibitory cells that peak around SSRT vs later? Maybe you can highlight the specific 'representative' neuron by highlighting it in Figure 2b.

- Same comment for Figure 3c. In fact, from looking at Figure 3b, I cannot tell which neuron 3c comes from. Maybe you can highlight the specific neuron in Figure 3b, like suggested for Figure 2b.

We now include additional example neurons for each type in the Main text figures. These examples were chosen to be complementary by illustrating neurons in different layers with different spike widths and modulation times. In the time-depth plots (panel b of each figure), we added an arrow identifying the example neurons.

Regarding Figure 3, although the lower panel of Fig 3a shows the recruitment of significant activity for the entire population, Fig 3b shows this only for sessions in which we could confidently assign neurons to layers. Therefore, not every neuron shown in panel a is also in panel b.

- In the first paragraph of the Discussion section the authors refer to Figure 6. However, I prefer the explanation of this figure to be embedded in the Discussion instead of having an entirely separate paragraph in the figure description. This will make it more of a complete single story, instead of having a separate story in the figure description.

Seeking to balance word count and clarity, we wrote the caption of Figure 6 to be self-sufficient. To address the reviewer's comment, we have now isolated the conjecture on microcircuitry, which overlaps with Figure 6, in a new section, **Cortical Microcircuitry of Executive Control (Page 17-20)**. We have also added some text from the figure 6 caption to the main text of the Discussion to ensure all the material in Figure 6 is also present there. We understand that making this change increases our word count.

Line edits

- Please have a native English speaker smooth out the manuscript. Some sentences read like notes rather than full sentences (e.g. line 53), and there are many minor mistakes that could be corrected, like missing articles (e.g. line 17, 40, 51, 606, 610).

As requested, we have revised the manuscript to improve the flow and correct grammar and spelling. Native English speakers have also reviewed the manuscript.

Reviewer #3 (Remarks to the Author):

This paper is an important step toward a better understanding of cortical microcircuitry involved in executive control. As a continued effort to dissect laminar organization of medial frontal cortex during stop-signal task (Sajad et al., Nat. Neuro. 2019), this paper focused on more critical trial types (canceled) and task epochs (around SSRT) and found interesting functional dissociations across different neuronal groups. In particular, it is noteworthy finding that “goal maintenance” narrow-spiking neurons are specifically localized in L2/3. Furthermore, the report of N2/N3 in the monkey model is novel and its relation to neural spiking activates is an important addition to the field. The paper has solid and convincing results. That said, I have some suggestions that can potentially improve the paper.

We thank the reviewer for their kind comments and appreciation of the manuscript.

- Major points

More justification is needed for the approach of classifying individual neurons into distinct groups – conflict monitoring, event timing, goal maintenance. While it is justified in terms of previous findings (e.g., for conflict neurons) and behavioral demands, this classification seems arbitrary, post-hoc, and not based on first principles (i.e., derived from theory or model). Why are there three, not two or not four, categories? Even clustering results in Supplementary figure 1 do not speak to this issue.

The reviewer is asking a core question. Our approach is motivated by the comprehensive understanding of the early visual pathway derived from characterizing and distinguishing neuron types (e.g., Rowe MH, Stone J. The interpretation of variation in the classification of nerve cells. Brain Behav Evol. 1980;17(2):123-51). We have revised the manuscript to explain how our approach emphasizes unbiased clustering methods and evaluates all neurons through converging constraints derived from the principles of quantitative models used to describe executive control (**Page 4, Lines 91-108**). **Figure 1** and associated supplementary figures were thoroughly revised to clarify our approach.

Another way to ask this question is, how should readers interpret the different number of neurons in each group? Does it matter? Because there is a relatively smaller number of goal-maintenance neurons, should we conclude that SEF is involved more in conflicting monitoring and time keeping? Or does SEF do everything including the processing of gain and loss (Sajad et al., Nat. Neuro. 2019)? How these different groups of neurons interact and what is the population-level understanding? I know this is a high-level question that may need brand-new analyses (like population-level decoding or state-space/dynamical-systems approach). But at least, this issue should be thoroughly discussed as medial frontal cortex has notoriously heterogeneous neural populations in their sensorimotor selectivity.

We appreciate these questions because we ask them ourselves. Here we address the questions in the order they were asked: We address the first question about sample numbers on **page 21 of Discussion**, in the first paragraph of a new section called ***Incidence and Multiplexing of Signals***. In short, we believe that conclusions derived from relative sampling frequencies are too risky given the uncertainties inherent in extracellular spike sampling.

We address the question about state-space/dynamical systems approach also in that section (**Page 21, Lines 654-671**). We are also describing how signals reported in this study multiplexed with those reported before in Sajad et al 2019. (**Page 7, Lines 172-179; Page 9, Lines 257-262; and Page 11-12, Lines 335-339**). In short, the population-level state-space/dynamical-systems analysis methods treat neurons as equivalent, but decades of previous research have demonstrated that functional differences among neurons within and across layers correspond to differences in morphology, biophysics, and connectivity. Therefore, we believe a catalogue of functional differences is most informative now. Certainly, future research can employ alternative approaches to investigate these data.

Another key question is, is this classification a distinct category or more like a continuum? The latter is more likely given their overlapped dynamics (Fig. 5b, Sup. Fig. 1e). The difference between conflict and maintenance neurons is

only about how long the neurons sustain their activities (e.g., transient versus persistent) and so not clear cut at all. Specifically, maintain neurons can be well considered as conflict monitoring and vice versa. This issue seems to become more problematic as even predictions between models are similar – as mentioned in methods, model predictions and parameters are correlated (e.g., SSD, p(error), conflict, and time to tone) and it is not clear how “random variations” (across single trials?) can differentiate these models.

The reviewer raises two important points:

- 1) Are categories discrete or continuous? Indeed, neuron classification has always been a non-trivial problem in neurophysiology (see Rowe and Stone citation above). Even in sensorimotor areas with robust sensory- and motor-related responses the distinction between sensory and motor neurons has proven to be a challenge. But many studies draw the line based on objective criteria, acknowledging that there can be a gray zone. In this study, we have drawn the line between the two facilitated neuronal classes based on objective criteria as well. We acknowledge that there is a gray zone and there can be some degree of misclassification, but these two populations exhibited other differences beyond their duration that justifies their separation into two distinct neuronal populations. We have now emphasized these differences in the **Results (Page 10, Lines 285-287; Lines 294-296; Page 10-11, Lines 302-304; Page 12, Lines 350-351)**. We have also added a new **Supplementary Figure 6e** that illustrates that Goal Maintenance and Conflict neurons exhibited an opposite overall relationship to performance parameters.
- 2) Are the model tests decisive? We have addressed this in **Methods (Page 28, Lines 886-890)** and have plotted the degree of difference between the tested models in our new **Supplementary Figure 3b**. In short, the association of the different neuron modulation patterns with different models can be determined with Bayesian reliability because task performance and derived behavioral parameters varied across sessions.

Along the line of the previous comment, model comparison results should be presented better. In particular, details of alternative models are buried in the text/supplementary and hard to digest. It'd be useful to have a figure showing predictions from different models (well explained in method, “mixed-effects model”) and to bring a simplified version of Sup. Table 2 into the main manuscript.

We have revised the manuscript substantially based on this suggestion. We moved a substantial portion of Supplementary Figure 2 to **Figure 1** to explain the models. In **Figures 2, 3, 4, and 5** we have moved from previous Supplementary Table 2 the panels showing the model-comparison results. We have clearly defined the models based on their mathematical descriptions and provided a summary for our nomenclature in our new **Supplementary Table 2**. We have added a new **Supplementary Figure 3a** explaining different hazard rate models in more detail. We have added a new **Supplementary Figure 3b** showing how parameters related to different models differ from each other. We have revised the text in the **Results section** to orient the reader to the different models with simpler language (**Page 5-6, Lines 99-145**). Finally, when reporting model comparisons in **Results**, we report which model was the best and remind the reader which models were excluded.

Another big question is, what is the behavioral relevance of these activities in stop-signal task. More specifically, the behavioral relevance of conflict monitoring is not directly tested. In discussion, it is mentioned that conflict monitoring can be useful for post-stopping slowing but this interesting idea is not tested at all. For instance, was the go RT slower in the next trial after the conflict-monitoring neurons showed a higher firing rate? Does animal use timing information in ‘time-keeping neurons’ to predict SSD and adjust behavior accordingly?

We thank the reviewer for this important question. We now describe the relationships of the neural signals to adjustments in RT on **Page 6-7, Lines 167-170, Page 9, Lines 249-255, and Page 11, Lines 331-333**, with the Methods (**Page 28, Lines 904 - 908**). We also report in

Results (page 11, Lines 328-330) and show in **Supplementary Figure 6f** that the activity of Goal Maintenance neurons was lower when monkeys aborted canceled trials.

Given that no specificity was found for conflict neurons in terms of laminar and spiking width (χ^2 analyses), the conclusion that those neurons are broad-spiking and in L3/5 (in abstract, Line 23-25) is not warranted. This is just a by-product of sampling neurons in SEF. The valid conclusion is that no specificity was found.

We thank the reviewer for this astute comment and have revised the text in the **Abstract**. We would note here that our statistical analysis for the laminar distribution of neurons does account for the sampling bias (**Supplementary Table 1**). Also, we trust that the reviewer agrees with us that the interpretability and importance of the results do not depend on the laminar distribution being different from the sampling distribution.

Motivation for classifying neurons into broad-spiking and narrow-spiking is also not clear. It would be useful to provide backgrounds or hypotheses in the introduction. For example, is there any previous anatomical study suggesting inhibitory interneurons are more common in L2/3? It seems to be the case in Supp. Fig. 1.

We have clarified the motivation in **Results** where we first report broad- vs. narrow-spiking neurons (**Page 7, Lines 181-184**). When describing the spiking widths of the other neuron types, we now state that this is to inform the contribution of neurons within the microcircuitry. We trust that the reviewer agrees that this information is useful to constrain microcircuit models that can enact different functions (e.g., Michael X Cohen, *Trends in Neuroscience*, 2014).

It'd be useful to explicitly state the null hypothesis when linking N2/P3 and spiking activities across layers. Is it somewhat obvious that ERPs would be better predicted by upper layers as they are closer? Also, due to their match in timing, isn't it obvious that monitoring/timing neurons better predict N2 and maintenance neurons better predict P3? One way to tackle the former question is to test how well the cranial EEG is predicted by LFP signals across layers.

We have addressed this in **Discussion (Page 20-21, Line 631-644)**.

- Minor points

In figure 2,3, and 4, the bottom plots in panel (a) have a confusing label, P(active). Because p(active) is different between left and right panels, it'd be better to use a separate y-axis label.

We appreciate this suggestion and have edited the figures.

I can easily imagine, to readers naïve to the stop-signal task, the paper is written a bit difficult to follow and jargon-heavy fashion. If it is written without assuming prior knowledge, the paper will receive a wider readership.

We have revised the text throughout. Major changes were made in the beginning of **Results** where we introduce the task and models (**Page 3, Line 52-62; Page 4-6, Lines 94 - 143**), explaining the testing procedures in the main text (**Results, Figure 1**), and revising the Supplementary material to provide a more thorough explanation of concepts relevant to our analyses (**Supplementary Figures 1-3**). We are anxious to learn if the reviewer finds the manuscript more accessible to the general reader.

Y-axis Label of the right panel in fig. 3c should be pre-SSRT, to be consistent with left panel shades and legends.

Thank you. We have now corrected this in the new panel **Figure 3e**

Why did conflict neurons show higher activity right before feedback tone (Fig. 2a, Sup. Fig. 3b)?

We are not sure whether the Reviewer is asking about the activity on no-stop trials that appear on panels 2a and Supplementary Fig 3b or is asking about the activity on canceled trials. We address both in turn:

- 1) The activity of neurons aligned on tone, on no-stop trials (figure 2a, and Sup. Fig 3b) during the pre-tone interval can be larger than that on Canceled trials simply because of post-saccadic activity bleeding into this time period. Therefore, this appearance of modulation is incidental to other effects. We have now added this in the figure captions (e.g., **Fig 2a** and **Supplementary Figure 4b**).
- 2) With respect to the activity of Conflict neurons on Canceled trials prior to tone, it was observed as a weak modulation in a low proportion of neurons. We were not confident proposing a definite interpretation, but we included the right panel in Fig 2b for transparency and consistency with the other figures. The most conservative interpretation must parallel that of the Goal Maintenance neurons. But as we have explained in the manuscript there is some degree of multiplexing that can also explain this.

REVIEWER COMMENTS

Reviewer #1 (Remarks to the Author):

The authors have adequately addressed my questions and I am happy to recommend this manuscript for publication.

Reviewer #2 (Remarks to the Author):

We thank the authors for their diligent efforts to improve the quality and readability of their manuscript.

We have one final point, which is about this comment:

"Line 876-878. It is unclear why the set of SSDs for the different monkeys have different separation sizes (40-60 ms vs. 100 ms). This needs explanation."

Your reply: "We have explained this in the revised Methods section and cited relevant literature on Page 25, Lines 797-798; Page 3, Line 57-58. The key aspect of the stop-signal task is to have a set of SSDs that probe response inhibition low to high error rates. Monkeys, like people, are idiosyncratic, so different SSD values must be used."

The authors misunderstood our comment. The question was why the experimental manipulation differed between the two monkeys? We do understand the SSD titration procedure, and why it's needed, but we missed why the resolution of the titration for one monkey was twice that of the other monkey. Please clarify.

Reviewer #3 (Remarks to the Author):

I appreciate the authors' thorough response to the points raised by me and the other reviewers. More specifically, the paper is significantly improved by the unbiased clustering of neuronal types, better presentation of model comparisons, and also written better to address the reviewer's concerns. I fully support publication.

I just have two minor suggestions: 1) I do not see panel f in supplementary figure 6, which might be lost due to technical issues during editing. 2) For the time-keeping neuron (figure 3), isn't it much more straightforward to present data from the time of Go signal (i.e., as a function of SSD)? I understand the current plot ("time from SSRT") is more consistent with how the other types of neurons are shown but I expect readers would wonder why activities in figure 3a reflect time-keeping. If the activities are plotted as a function of time from Go, time-keeping/ramping in those neurons would be much more clearly shown as their main feature.

Sajad Errington Schall – Response to reviews

REVIEWERS' COMMENTS

Reviewer #1 (Remarks to the Author):

The authors have adequately addressed my questions and I am happy to recommend this manuscript for publication.

We are pleased to have satisfied the reviewer and appreciate the helpful comments.

Reviewer #2 (Remarks to the Author):

We thank the authors for their diligent efforts to improve the quality and readability of their manuscript.

We are pleased to have satisfied the reviewer and appreciate the helpful comments.

We have one final point, which is about this comment:

"Line 876-878. It is unclear why the set of SSDs for the different monkeys have different separation sizes (40-60 ms vs. 100 ms). This needs explanation."

Your reply: "We have explained this in the revised Methods section and cited relevant literature on Page 25, Lines 797-798; Page 3, Line 57-58. The key aspect of the stop-signal task is to have a set of SSDs that probe response inhibition low to high error rates. Monkeys, like people, are idiosyncratic, so different SSD values must be used."

The authors misunderstood our comment. The question was why the experimental manipulation differed between the two monkeys? We do understand the SSD titration procedure, and why it's needed, but we missed why the resolution of the titration for one monkey was twice that of the other monkey. Please clarify.

We apologize for the misunderstanding.

The revised text reads, "The selection of SSDs was adjusted to the idiosyncrasies of each subject to ensure performance satisfying key criteria for stop-signal task. Different SSD values were used for the two subjects to account for between-subject differences in stopping performance ¹."

In running various participants (human or monkey) on this task adjustments of SSD parameters to account for individual differences is common. The key criteria satisfied by all participants is successful stopping in 50% of stop signal trials and $RT_{\text{noncanceled}} < RT_{\text{no-stop-signal}}$. Now, we have not addressed why monkeys X and Eu were different because we do not have enough information. The difference is most likely derived from the incident different training histories of the two monkeys. If the reviewer wishes us to expand on this, we are happy to, but every macaque

behavior neurophysiology study includes idiosyncratic differences like this.

Reviewer #3 (Remarks to the Author):

I appreciate the authors' thorough response to the points raised by me and the other reviewers. More specifically, the paper is significantly improved by the unbiased clustering of neuronal types, better presentation of model comparisons, and also written better to address the reviewer's concerns. I fully support publication.

I just have two minor suggestions: 1) I do not see panel f in supplementary figure 6, which might be lost due to technical issues during editing.

We apologize. Panel f is now included in Supplementary Fig 6.

2) For the time-keeping neuron (figure 3), isn't it much more straightforward to present data from the time of Go signal (i.e., as a function of SSD)? I understand the current plot ("time from SSRT") is more consistent with how the other types of neurons are shown but I expect readers would wonder why activities in figure 3a reflect time-keeping. If the activities are plotted as a function of time from Go, time-keeping/ramping in those neurons would be much more clearly shown as their main feature.

We agree with the reviewer about the advantages of the suggested illustration. However, such an alignment would obscure the pronounced suppression of these neurons after SSRT. To address this comment, though, the revised figure now explicitly marks when the GO signal appeared. We would also note that interested readers can construct such a plot easily from the code and data deposited in the OSF repository.